# Constitutive activation and oncogenicity are mediated by loss of helical structure at the cytosolic boundary of thrombopoietin receptor mutant dimers

Jean-Philippe Defour[1,2†], Emilie Leroy[1,2†], Sharmila Dass[3†], Thomas Balligand[1,2], Gabriel Levy[1,2,4], Ian C Brett[3], Nicolas Papadopoulos[1,2,4], Céline Mouton[1,2,4], Lidvine Genet[1,2,4], Christian Pecquet[1,2,4], Judith Staerk[1,2,4], Steven O Smith[3*], Stefan N Constantinescu[1,2,4,5*]

[1]Ludwig Institute for Cancer Research, Brussels, Belgium; [2]de Duve Institute, Université catholique de Louvain, Brussels, Belgium; [3]Department of Biochemistry and Cell Biology, Stony Brook University, Stony Brook, NY, Newyork, United States; [4]WEL Research Institute, WELBIO Department, Wavre, Belgium; [5]Ludwig Institute for Cancer Research, Nuffield Department of Medicine, Oxford University, Oxford, United Kingdom

**\*For correspondence:**
steven.o.smith@stonybrook.edu (SOS);
stefan.constantinescu@bru.licr.org (SNC)

†These authors contributed equally to this work

**Abstract** Dimerization of the thrombopoietin receptor (TpoR) is necessary for receptor activation and downstream signaling through activated Janus kinase 2. We have shown previously that different orientations of the transmembrane (TM) helices within a receptor dimer can lead to different signaling outputs. Here we addressed the structural basis of activation for receptor mutations S505N and W515K that induce myeloproliferative neoplasms. We show using *in vivo* bone marrow reconstitution experiments that ligand-independent activation of TpoR by TM asparagine (Asn) substitutions is proportional to the proximity of the Asn mutation to the intracellular membrane surface. Solid-state NMR experiments on TM peptides indicate a progressive loss of helical structure in the juxtamembrane (JM) R/KWQFP motif with proximity of Asn substitutions to the cytosolic boundary. Mutational studies in the TpoR cytosolic JM region show that loss of the helical structure in the JM motif by itself can induce activation, but only when localized to a maximum of six amino acids downstream of W515, the helicity of the remaining region until Box 1 being required for receptor function. The constitutive activation of TpoR mutants S505N and W515K can be inhibited by rotation of TM helices within the TpoR dimer, which also restores helicity around W515. Together, these data allow us to develop a general model for activation of TpoR and explain the critical role of the JM W515 residue in the regulation of the activity of the receptor.

## Editor's evaluation

The work provides a nice structure-function study providing clues as to how helix orientation can control transmission of signals across cell membranes. Moreover, the approach may find use in further studies exploring similar signalling involving other related membrane systems.

## Introduction

Cytokine receptors regulate key functions in the human body, such as growth and maintenance of cells in blood and the immune system (*Robb, 2007*). Their activation induces long-lasting genetic

effects by activation of transcription factors via their pre-bound cytosolic Janus kinases. The thrombo-poietin (Tpo) receptor (c-Mpl or TpoR) and its ligand Tpo are the main regulators of megakaryocyte differentiation and platelet production (*Kaushansky et al., 1994*). Besides these regulatory roles, TpoR functions in early hematopoiesis by promoting self-renewal, expansion, and maintenance of the hematopoietic stem cell pool (*Qian et al., 2007*; *Solar et al., 1998*; *Yoshihara et al., 2007*; *Fox et al., 2002*).

TpoR is a type 1 member of the cytokine receptor superfamily that also includes the erythropoi-etin receptor (EpoR), growth hormone receptor (GHR), granulocyte-colony stimulating factor receptor (G-CSFR), and prolactin receptor (PrlR). These receptors consist of two identical chains and function as homodimers (*Watowich et al., 1996*). They almost exclusively employ the Janus kinase JAK2, which is prebound to the cytosolic domain of the receptor. JAK2 contains both an activating kinase domain (JH1) and a pseudokinase domain (JH2). The JH2 domain exerts inhibitory effects on the JH1 kinase domain and is required for cytokine-induced activation of the kinase domain (*Bandaranayake et al., 2012*; *Ungureanu et al., 2011*; *Saharinen et al., 2000*). Upon activation, Y1007 and Y1008 in the kinase activation loop are trans-phosphorylated, which switches on the kinase activity and allows phosphorylation of other sites (*Feng et al., 1997*).

A fraction of EpoR (*Constantinescu et al., 2001b*), GHR (*Yang et al., 2007*; *Brooks et al., 2014*), gp130 (*Tenhumberg et al., 2006*), and PrlR (*Qazi et al., 2006*) is believed to exist as ligand-independent preformed dimers. Ligand binding induces a conformational change in the receptor that is coupled to a change in the orientation and possibly structure of the cytosolic juxtamembrane (JM) domain and which is transmitted to the intracellular JAK2 proteins. The mechanism of this coupling is at the heart of how these receptors transduce cellular signals. For example, *Brooks et al., 2014* have proposed a mechanism of GHR activation involving a transition of the dimeric transmembrane (TM) domain from a parallel orientation of the two TM helices to a left-handed coiled coil geometry that results in the separation of the inhibitory JH2 domain of one JAK2 from the JH1 kinase domain of the other. These studies emphasize the importance of the rotational orientation and tilt angles of TM helices in transmitting signals across cell membranes. The relative orientation and tilt of TM helices are often modulated by extracellular signals in polytopic membrane proteins (*Hall et al., 2011*; *Ren et al., 2016*) and in a fashion similar to the cytokine receptors, the rotational orientation of the single pass receptor tyrosine kinases controls receptor signaling by altering the structure and membrane interactions of the juxtamembrane sequence (*Bell et al., 2000*; *McLaughlin et al., 2005*).

The human TpoR (hTpoR) differs from the other type 1 cytokine receptors in several respects. First, the TM domains of hTpoR do not readily dimerize in an inactive receptor (*Defour et al., 2013*). Rather, ligand binding induces dimerization of the receptor in an active conformation (*Wilmes et al., 2020*). We have engineered receptor dimers of TpoR using either the dimerization domain of Put3 in the murine TpoR (mTpoR) (*Staerk et al., 2011*) or by specific asparagine (Asn) substitutions within the TM domain (*Leroy et al., 2016*). Results from the engineered Put3 coiled coil dimers suggest there is one inactive orientation of the TM helices and several orientations that lead to different signaling outputs. The observation of active and inactive orientations of the hTpoR TM helices has also been reported with the use of cysteine cross-linking and alanine-scanning mutagenesis (*Matthews et al., 2011*). More recently, we have explored the use of Asn-scanning mutagenesis to modulate the orientation of the TM helices (*Leroy et al., 2016*). Asn induces hydrogen bonding between the Asn residues (*Choma et al., 2000*; *Zhou et al., 2000*) and dimerization of cytokine type I receptors TMD, as TpoR, EpoR or G-CSFR, leading to signaling in the absence of ligand (*Ding et al., 2009*; *Maxson et al., 2016*; *Becker et al., 2008*). The striking observation was that when Asn residues are introduced by substi-tution at different positions, the TM helices in the murine receptor form dimers in various azimuthal orientations, whereas only a single substitution in the human receptor (S505N) results in appreciable activity. The S505N mutation is found in familial forms (*Ding et al., 2004*) and in rare sporadic cases of essential thrombocythemia (ET) and primary myelofibrosis (PMF) (*Beer et al., 2008*; *Ma et al., 2011*).

A second major difference between TpoR and other type 1 cytokine receptors is the presence of a five-residue (K/RWQFP respectively in m/hTpoR) insertion at the cytosolic JM boundary. The most prevalent hTpoR mutations in myeloproliferative neoplasms are within this motif at W515 within this motif (W515L/K/A/R) (*Beer et al., 2008*; *Pardanani et al., 2006*; *Pecquet et al., 2010*; *Pikman et al., 2006*). We have shown that W515 and the surrounding residues are required to prevent self-activation of hTpoR as W515 prevents productive dimerization of the upstream TM helices (*Defour et al., 2013*;

*Staerk et al., 2006*). Strikingly, 17 out of the 20 possible amino acids activate hTpoR when introduced at W515, including W515Y/F (*Defour et al., 2016*), indicating a very specific physiological effect of the tryptophan (Trp) indole side chain in maintaining the inactive state of hTpoR.

In this article, we take advantage of the Asn and Trp mutations as tools to address the mechanism of TpoR activation. We use the Asn mutations to explore how dimerization of the TM helices in a specific orientation may trigger activation, and we use the Trp mutations to interrogate how the intracellular JM sequence couples the orientation of the TM domains to the orientation and proximity of the JAK2 JH1 and JH2 domains. We ask: what is the structural basis of activation of JAK2 by the TM and JM mutants of TpoR, and how can this activation be turned off? A combination of solid-state NMR spectroscopy, mutagenesis, biochemical assays, and *in vivo* bone marrow transplantation is used to probe structural changes induced by mutations and then to validate the structural data in live cells and *in vivo*.

## Results and discussion

### Ligand-independent activity of Asn mutants is dependent on both the rotational orientation and the proximity of the mutation to the cytosolic membrane surface

Asn substitutions drive dimerization of TpoR TM domains and induce very different effects in the murine and human TpoRs (*Leroy et al., 2016*). In the murine receptor, there are several TM positions where Asn substitutions result in constitutive activity (*Figure 1*). The data reveal that there is primarily a single position around each helical turn that has increased activity, namely V494N, S498N, and G502N. Moreover, measurements of Ba/F3 cell proliferation with increasing concentrations of the Tpo ligand show that there is increasing autonomous cell growth in the order V494N<S498N<G502N (*Figure 1d*), which is in agreement with our previous study (*Leroy et al., 2016*).

V494, S498, and G502 lie on one face of the mTpoR TM helix (*Figure 1b*). We have previously shown that the murine receptor forms an inactive dimer mediated by TM helix interactions and that the inactive interface is centered on A499 (*Leroy et al., 2016*). In contrast to the hTpoR, there appears to be several active interfaces for the mTpoR (*Staerk et al., 2011*; *Matthews et al., 2011*), which in principle would be contrary to a very rigid TM-JM region and would suggest a certain level of flexibility. It is important to note that Asn mutants on one specific dimeric interface generate an increase in activity proportional to proximity to the cytosolic end of the TM domain (*Figure 1c and d*). For example, L495 is two helical turns above G502, yet the L495N activity is comparable to that of an inactive receptor. The same increase of activity closer to the cytoplasmic boundary is observed when comparing L493N and L500N even though the L493-L500 interface is oriented 180° from the strongly active G502-S498 interface.

### *In vivo* effects of the TpoR V494N, S498N, and G502N mutants

Before undertaking experiments to probe the rotational and longitudinal dependence of the Asn mutants, we first validated the cell growth measurements obtained in *Figure 1d* with stably transduced Ba/F3 cells using *in vivo* bone marrow reconstitution experiments and bone marrow cells. In these experiments, bone marrow cells from C57BL/6J mice were transduced with bicistronic retroviruses coding for the V494N, S498N, and G502N mTpoR mutants or wild type (WT) mTpoR and infected cells were used to transplant lethally irradiated C57BL/6J mice. *Figure 2a* and *Figure 2— figure supplement 1a* show that 40 d after reconstitution a gradient of *in vivo* effects could be detected, with strong, mild, and weak signs of myeloproliferation for G502N, S498N, and V494N mutants, respectively. Bone marrow histology and spleen histology showed fibrosis for the spleens and bone marrow samples of G502N mice (*Figure 2b*), with an increase in granulocytes but no erythrocytosis reminiscent of an early primary myelofibrosis (pre-PMF) phenotype where the myeloid to erythroid ratio is increased (*Swerdlow et al., 2017*). Furthermore, the weights of both spleen and liver were higher for G502N than for S498N, and at the high limit of normal for V494N (*Figure 2c*). Survival curves were in line with the severity of the disease induced by the mutants (*Figure 2—figure supplement 1b*). The *in vivo* myeloproliferation phenotype detected by hematological parameters and by spleen and liver histology correlated with the measured levels of TpoR activity.

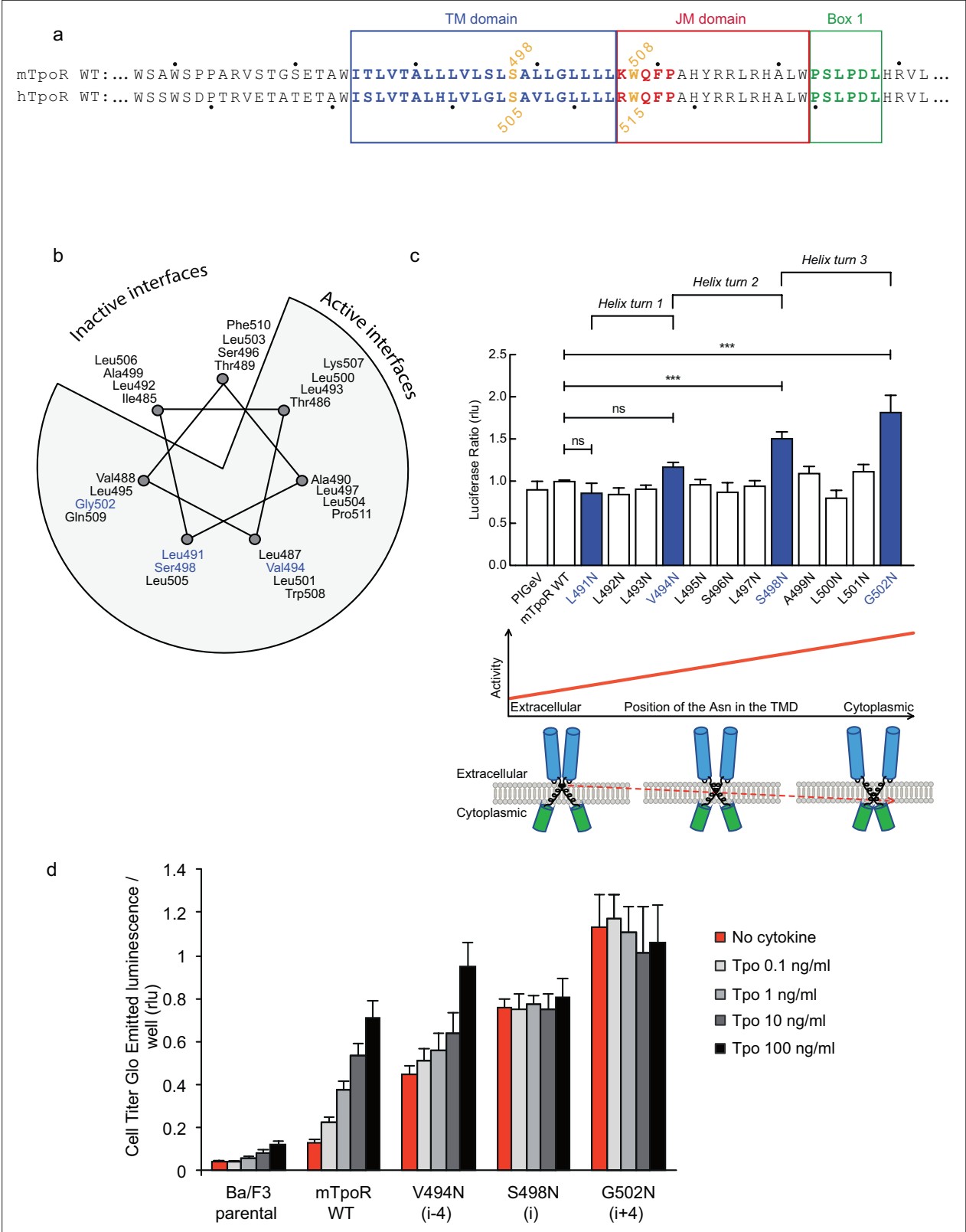

**Figure 1.** Activity and relative position of asparagine (Asn) mutants in murine thrombopoietin receptor (TpoR). The murine TpoR (mTpoR) wild type (WT) transmembrane domain sequence and numbering. The residues that make the KWQFP motif are highlighted in red (**a**). A helical wheel diagram shows the positions of L491, V494, S498, and G502 (blue) on the active interface (**b**). The relative tranlscriptional activity of mTpoR WT or mutants was assessed with STAT5 reporter Spi-Luc in HEK-293T cells. Shown are averages of separate experiments ± SEM (n = 3–4), each experiment being performed with

*Figure 1 continued on next page*

*Figure 1 continued*

three biological repeats for each condition (triplicates). Kruskal–Wallis non-parametric test with multiple-comparisons using Steel's test with mTpoR WT as control (jmp pro12). \*\*\*p<0.0001; n.s., nonsignificant (**c**, upper panel). Cartoon illustrating how asparagine residues are predicted to promote crossing of transmembrane helices (**c**, lower panels). Short-term cell growth was measured using the Cell-Titer-Glo luminescent cell viability assay (Cell Titer Glo, Promega) with stable Ba/F3 cell lines expressing mTpoR WT or mutants with no ligand or increasing concentrations of Tpo ligand (0.1, 1, 10, and 100 ng/ml). Shown are the averages of three independent experiments ± SD (each experiment performed once) (**d**).

The online version of this article includes the following source data for figure 1:

**Source data 1.** Raw data, scatter plot, and statistics (Prism 9.1.2, jmp pro12) for *Figure 1c* (upper panel).

**Source data 2.** Raw data for *Figure 1d*.

## Asn mutations near the cytoplasmic boundary induce local changes in secondary structure of the K/RWQFP motif

The results above raise the question of how the rotational and longitudinal positions of the Asn substitutions are coupled to receptor activation. To address this mechanism, we took advantage of TM

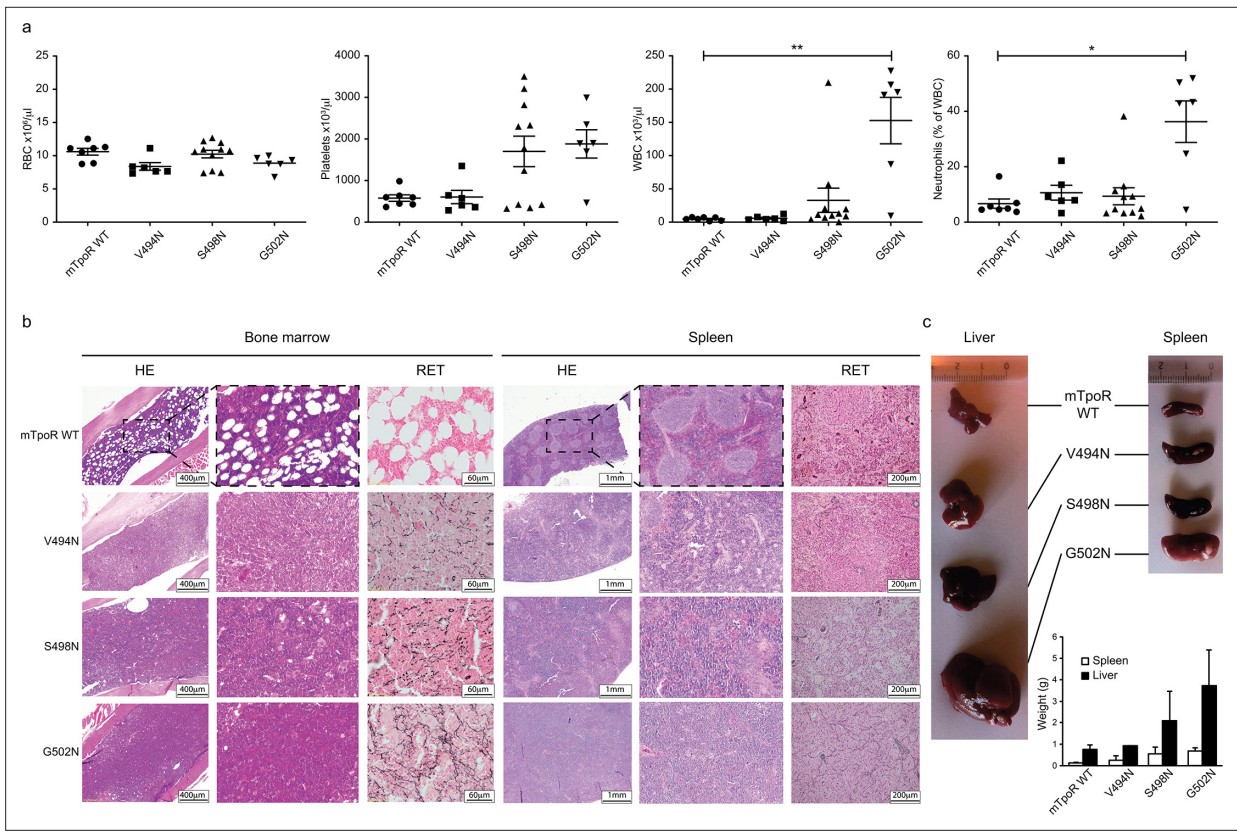

**Figure 2.** *In vivo* data comparing asparagine (Asn) murine thrombopoietin receptor (TpoR) mutants. Red blood cell (RBC), platelets, white blood cells (WBC), and neutrophils were measured using a MS9 blood cell counter at day 40 post-transplantation. Values shown represent the average of at least six biological replicates ± SEM., Kruskal–Wallis nonparametric test with Dunn's multiple-comparisons test (Prism 9.1.2); \*p<0.05, \*\*p<0.01 (**a**). Bone marrow/spleen cellularity and fibrosis have been evaluated by hematoxylin-eosin (HE) and reticulin (RET) staining, respectively. 3× zoom was performed on the .jpg file with Adobe Illustrator. (**b**). Liver and spleen were photographed and weighted (**c**). Values shown represent the average of two biological replicates ± SEM.

The online version of this article includes the following source data and figure supplement(s) for figure 2:

**Source data 1.** Raw data and statistics (Prism 9.1.2) for *Figure 2a*.

**Source data 2.** Raw data for *Figure 2c*.

**Figure supplement 1.** Mouse models expressing V494N, S498N, and G502N mutants of thrombopoietin receptor (TpoR) show weak, mild, and strong signs of myeloproliferation with a significant reduction of survival in the case of G502N mutants.

**Figure supplement 1—source data 1.** Raw data and statistics (Prism 9.1.2) for *Figure 2—figure supplement 1b*.

peptides containing the regulatory JM K/RWQFP motif that is unique to the TpoR. Structurally, the TpoR TM domain can fold into a membrane spanning α-helix independent of the rest of the receptor domains and the effects of mutations within the TM-JM peptide sequence parallel the effects observed upon mutation of the full receptor (*Defour et al., 2013*). For example, the TM-JM peptide sequence corresponding to hTpoR is largely monomeric but is driven into an active dimer conformation with the S505N and W515K mutations (*Defour et al., 2013*). In this section, we use solid-state NMR measurements of mTpoR TM peptides to address the structure of the peptide at the boundary between the hydrophobic TM domain and the KWQFP sequence with the aim of understanding how dimerization induced by the TM S498N and G502N mutations is coupled to structural changes in the JM sequence. We have previously shown that the TM-JM peptides reconstituted into model membrane bilayers composed of DMPC:DMPG are able to replicate the dimerization behavior of the full TpoR wild type and mutant receptors (*Defour et al., 2013*). The negatively charged DMPG provides a net negative charge to the membrane surface that mimics inner bilayer surface of native plasma membranes (see 'Materials and methods'). The proximity of these mutations to the K/RWQFP motif suggests that they may impact the structure near W508 in mTpoR or W515 in hTpoR, where many mutations trigger activation (*Defour et al., 2016*). We then address the secondary structure of the dimeric hTpoR TM-JM peptide containing the activating W515K mutation.

To gain insight into the secondary structure of TpoR TM-JM peptides, we combined solid-state NMR and FTIR. We performed solid-state NMR spectroscopy using peptides containing $^{13}$C-labeled amino acids at specific positions. Assignments were aided by FTIR measurements of the amide I vibration, which reveal that the TM-JM peptides are predominantly α-helical when reconstituted into membrane bilayers (*Defour et al., 2013*) without contributions from β-strand or β-sheet secondary structure. It is more challenging to distinguish random coil from α-helix by FTIR due to overlap of the vibrational bands. However, on the basis of the $^{13}$C NMR chemical shift of backbone carbonyl groups, we have previously shown that the α-helix of the TM domain extends into the JM region until at least F510, which is adjacent to P511 of the KWQFP motif (*Staerk et al., 2006*). The $^{13}$C=O NMR chemical shift is sensitive to secondary structure and occurs at ~175 ppm or higher when the $^{13}$C-labeled amino acid is within an α-helix due to direct hydrogen bonding to the i-4 NH group (*Saitô et al., 1998*). *Figure 3* presents $^{13}$C NMR spectra of murine and human TpoR TM-JM peptides containing 1-$^{13}$C L505 or 1-$^{13}$C L512, respectively (*Figure 3a*). This specific $^{13}$C=O label provides a probe of intramolecular hydrogen bonding with the NH of murine W508 or of human W515. In the WT mTpoR peptide, the chemical shift of L505 is at 176.5 ppm (*Figure 3b*) consistent with α-helical structure at the boundary of the TM domain and the KWQFP motif. In contrast in the S498N and G502N mutants, where the W508 group is rotated into the interface, the $^{13}$C=O chemical shift moves to lower frequency and the $^{13}$C NMR resonance begins to broaden (*Figure 3c and d*). We interpret these changes as a local unraveling of the α-helix in the region between L505 and W508 of the mTpoR. Importantly, the upfield chemical shift and broadening are greater in the G502N mutant compared to the S498N mutant.

Since our data with active Asn mutants indicated secondary structure changes around W508 in mTpoR, we hypothesized that the active hTpoR W515 mutations also induce local loss of the α-helicity. To test this hypothesis, we performed solid-state NMR on TM-JM peptides corresponding to the hTpoR sequence. We compared the $^{13}$C=O chemical shift at position L512 (equivalent to L505 in mTpoR) in the WT and W515K peptides derived from hTpoR. In the WT hTpoR (*Figure 3e*), the 1-$^{13}$C NMR resonance of L512 is narrow with a $^{13}$C chemical shift of 177.1 ppm consistent with α-helical structure. In contrast, the 1-$^{13}$C L512 resonance shifts to lower frequency and broadens in the W515K mutant of the hTpoR TM-JM peptide (*Figure 3f*) similar to the G502N mutation in the mTpoR TM-JM peptide. These observations suggest that W515K/L/A/R mutations achieve receptor activation by inducing a local loss of -helicity of the TM helix at its boundary with the cytosolic RWQFP motif.

## Unfolding of the RWQF α-helical motif is a common mechanism of receptor activation

Our working model for the mechanism of activation in the wild-type or mutant receptors is that the RWQF motif is stabilized in the inactive state as an α-helix as a result of a cation-π interaction between R514 and W515. This interaction allows the RWQF sequence to partition into the more hydrophobic head-group region of the bilayer. Both Arg and Trp are over-represented at transmembrane–intracellular junction of TM helices (*von Heijne, 1992*), but whereas Arg prefers a water-accessible

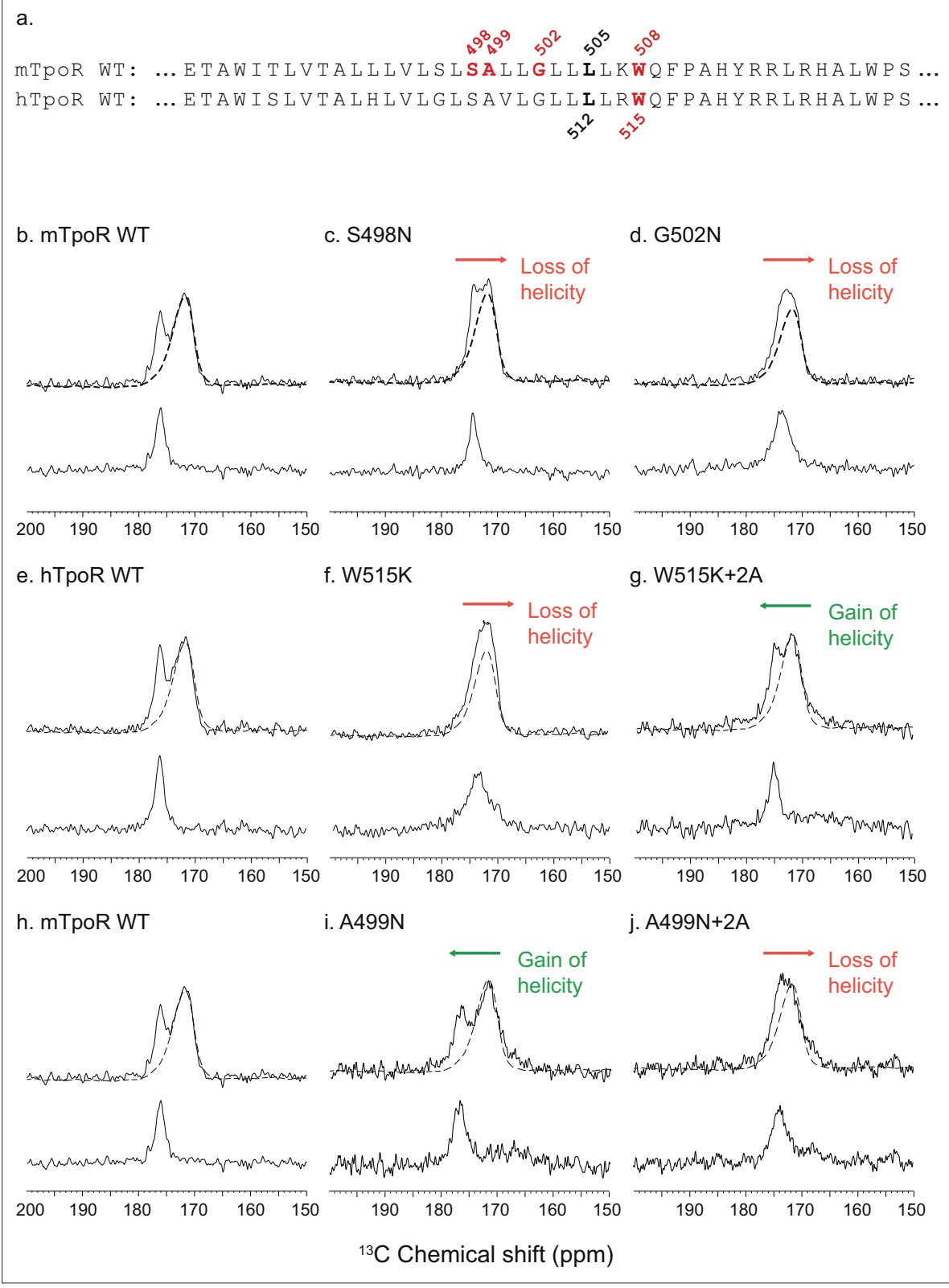

**Figure 3.** Secondary structure changes in the RWQFP insert upon activation. For solid-state NMR studies, peptides corresponding to the transmembrane and juxtamembrane (TM-JM) regions of thrombopoietin receptor (TpoR) were reconstituted into model membrane bilayers (**a**). 1D $^{13}$C MAS NMR spectra in the region of the $^{13}$C=O carbonyl resonances are shown of murine TpoR (mTpoR) wild type (WT) (**b**), mTpoR S498N (**c**), mTpoR G502N (**d**), human TpoR (hTpoR) WT (**e**), hTpoR W515K (**f**), hTpoR W515K+2 alanines (+2A) (**g**), mTpoR WT (**h**), mTpoR A499N (**i**), and mTpoR A499N+2A

*Figure 3 continued on next page*

*Figure 3 continued*

(**j**). The mTpoR peptides labeled with 1-[13]C L505 (murine) or 1-[13]C L512 (human) were reconstituted into DMPC:PG bilayers. A single MAS NMR spectrum of unlabeled mTpoR or hTpoR (dashed line) was obtained with a high signal:noise ratio for subtraction from the spectra of the reconstituted labeled peptides to obtain the resonance of the L505 (murine) or L512 (human) [13]C=O in the lower panels. For the reconstituted [13]C-labeled peptides, two independent technical replicates (reconstitutions and NMR data sets) were obtained for each experiment.

The online version of this article includes the following figure supplement(s) for figure 3:

**Figure supplement 1.** Solution NMR comparison of wild-type and W515K thrombopoietin receptor (TpoR) transmembrane peptides.

**Figure supplement 2.** AlphaFold predictions of thrombopoietin receptor (TpoR) W515X mutants.

**Figure supplement 3.** AlphaFold predictions of RRQFP, RKQFP, WRQFP, and WKQFP mutants.

environment, Trp prefers to be buried in a more hydrophobic environment (*Yau et al., 1998*). Since Arg precedes Trp in the sequence, partitioning into the membrane head-group region results in a favorable interaction of the positive charge associated with the guanidinium group of the R514 side chain with the partial negative charge associated with the aromatic surface of the W515 side chain. Partitioning of the RWQF sequence into the more water-inaccessible environment drives the formation of helical secondary structure as an unpaired backbone C=O...NH in a hydrophobic environment is estimated to cost 6 kcal/mol (*Engelman et al., 1986*).

In this model, activation of the receptor results in or is caused by disruption of the R514-W515 cation-π interaction. In the W515 mutants, R514 is no longer stabilized in a membrane environment and the helix containing the RWQFP sequence unravels to allow the positively charged side chain to reach outside of the membrane. In the case of the Asn mutants and in the wild-type receptor with bound Tpo, dimerization of hTpoR (or rotation of the TM helices in mTpoR dimer) places W515 in the center of the helix–helix interface. The data suggest that a steric clash of the W515 side chains results in unraveling of the cytoplasmic end of the TM helix.

Additional NMR data and computational data are provided in *Figure 3—figure supplements 1–3* to support the model of helix unraveling indicated by the solid-state NMR studies. Structurally, we have undertaken solution-NMR studies in sodium dodecylsulfate (SDS) of the wild-type hTpoR TM-JM peptide and its W515K mutant. Relaxation measurements of the backbone [15]N resonances show that W515K mutation leads to association of the TM helices (as observed in membrane bilayers), and that it induces upfield chemical shift changes in the RWQF sequence consistent with helix unraveling (*Figure 3—figure supplement 1*). Similar results were obtained for the S505N mutant of TpoR (481–520) (*Brett, 2012*). We had previously shown that both the W515K and S505N transmembrane sequences induce dimerization in membrane bilayers (DMPC:DMPG), as well as in detergent micelles (dodecylphosphocholine, DPC) (*Defour et al., 2013*). The solution NMR studies provide an independent probe of the structure of the region surrounding W515 upon helix dimerization. These studies show that the transmembrane domain of TpoR is helical in both the wild-type and mutant peptides and that dimerization induced by the W515K mutation results in unraveling of the helix in the RWQF insert region of the peptide.

Computationally, we used AlphaFold 2.0 (*Jumper et al., 2021*) calculations of hTpoR TM-JM peptides to predict the influence of all possible mutations at position 515 on the TM-JM helix structure. Remarkably, α-helix unraveling was predicted for 15 out of 20 possible amino acids at 515 (*Figure 3—figure supplement 2* and *Table 1*). Importantly, two of the mutations that are not predicted to cause helix unraveling are W515C and W515P. Experimentally, these two amino acid substitutions are the only ones that do not induce constitutive activity among all possible amino acid substitutions at W515 (*Defour et al., 2016*). Introducing a Trp at the preceding position 514 instead of R/K in W515K/R mutants reverses helix unfolding in AlphaFold simulations (*Figure 3—figure supplement 3* and *Table 2*). This agrees with our previous data that the WRQFP mutant is inactive and is essentially monomeric (*Defour et al., 2013*).

Overall, our data indicate that unraveling of the α-helix (i.e. helix to random coil transition associated with unraveling of the amide bond) around W515 is a common mechanism for both the Asn and W515 mutations that induce dimerization of TpoR. We further explore a possible mechanism below.

**Table 1.** AlphaFold prediction of human thrombopoietin receptor (TpoR) W515X mutants. Summary of the AlphaFold 2.0 (*Jumper et al., 2021*) predictions of human TpoR W515X. The simulations were performed on human TpoR residues 474–573 containing the transmembrane and juxta-membrane domains until Box 2. The absence or presence of unfolding around in the RWQFP motif is indicated, together with the position where loss of helicity occurs.

| TpoR mutation | Unfolding? | Start unfolding residue | Remarks |
|---|---|---|---|
| WT | No | NA | NA |
| W515A | Yes | Q516 | NA |
| W515C | No | NA | Kink increased at Q516 |
| W515D | Yes | Q516 | NA |
| W515E | No | NA | Kink increased at Q516 |
| W515F | Yes | Q516 | NA |
| W515G | Yes | G515 | Incomplete/partial unfolding |
| W515H | Yes | Q516 | NA |
| W515I | Yes | Q516 | NA |
| W515K | Yes | Q516 | NA |
| W515L | Yes | Q516 | NA |
| W515N | No | NA | Kink increased at Q516 |
| W515M | Yes | Q516 | NA |
| W515P | No | NA | New/higher kink at P515 |
| W515Q | No | NA | Kink increased at Q516 |
| W515R | Yes | Q516 | NA |
| W515S | Yes | Q516 | NA |
| W515T | Yes | | |
| W515V | Yes | Q516 | NA |
| W515Y | Yes | Q515 | NA |

## A localized but not extended JM loss of α-helicity is compatible with self-activation and Tpo-induced activation

The NMR data indicated that the activating mutations in the mTpoR (S498N, G502N) and hTpoR (W515K) TM-JM peptides induce a localized loss of α-helicity starting three residues before the K/RWQFP motif at position L505 in the region of transition between the TM domain and the K/RWQFP motif. A major question is whether this loss of secondary structure is localized or is more extensive, providing flexibility that might prompt large conformational changes and possibly allowing significant changes in the configuration of the appended JAK2, as suggested for the growth hormone receptor (*Brooks et al., 2014*). We undertook a mutagenesis approach where we substituted two consecutive amino acids in the intracellular JM domain, between the TM domain and Box 1, either to Gly-Pro or to Pro-Pro. These double substitutions are expected to disrupt helical secondary structure (*Figure 4a*), which also comes out in AlphaFold2 predictive model (*Figure 4b*; *Jumper et al., 2021*; *Varadi et al., 2022*).

**Table 2.** Characteristics of mutants of the RWQFP motif.
Summary of the characteristics of mutants of the RWQFP motif displayed in *Figure 3—figure supplement 3*.

| TM-JM motif | Unfolding? | Constitutive activity? |
|---|---|---|
| RWQFP (WT) | No | No |
| RKQFP | Yes | Yes |
| RRQFP | Yes | Yes |
| WRQFP | No | No |
| WKQFP | No | No |

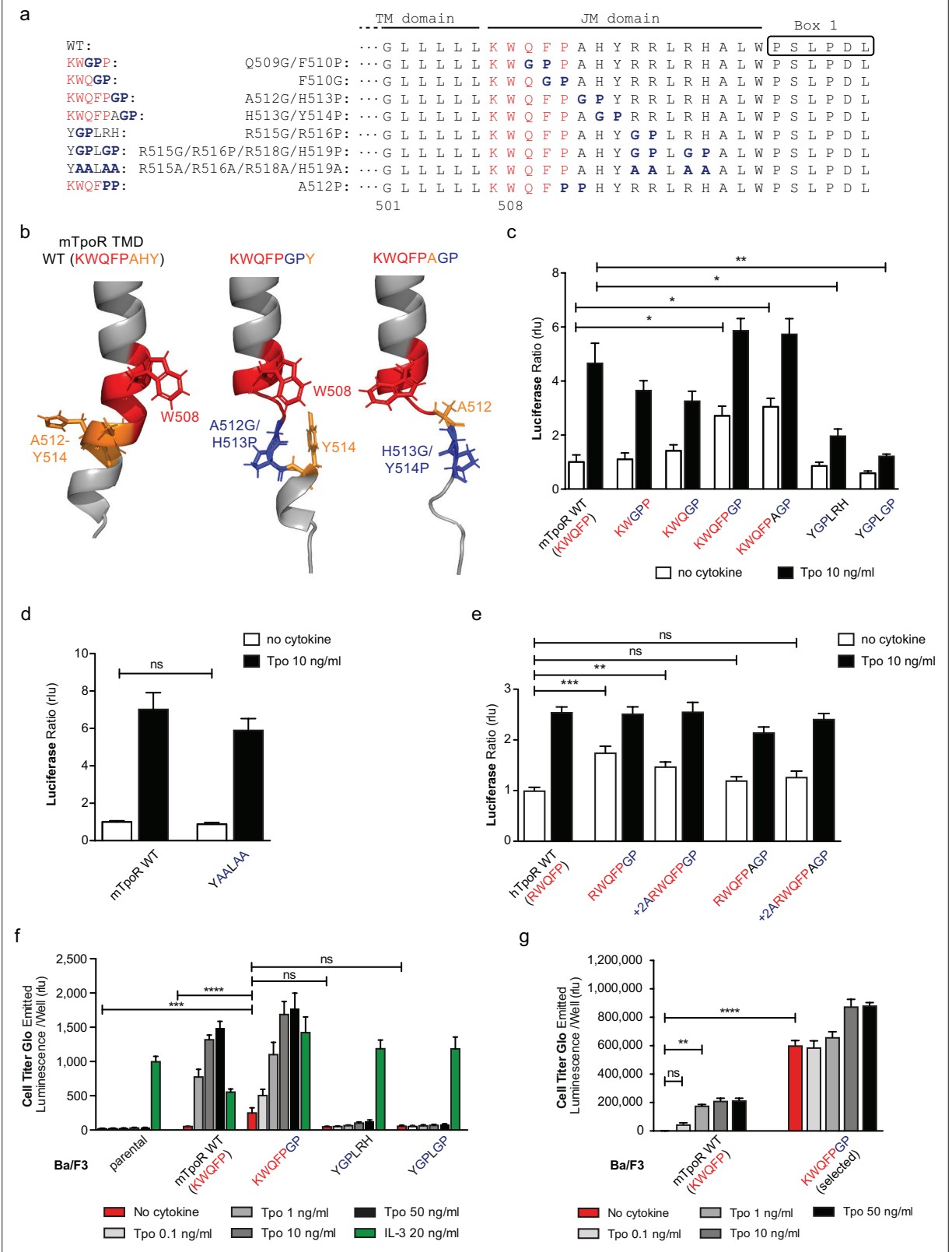

**Figure 4.** Localized substitution-induced loss of α-helicity of cytosolic juxtamembrane helix activates thrombopoietin receptor (TpoR) signaling in the absence of Tpo. Sequences of transmembrane (TM) and juxtamembrane (JM) sequences of murine TpoR (mTpoR) and position of substitutions to Gly-Pro, which interrupt secondary helical structure, and Ala substitutions, which are predicted to maintain secondary structure (**a**). The human TpoR (hTpoR) numbering adds +7 in the TM domain. For example, W515 in hTpoR is W508 in mTpoR. Models generated with AlphaFold2 (*Jumper et al., 2021*;

*Figure 4 continued on next page*

*Figure 4 continued*

**Varadi et al., 2022**) of the murine TM-JM domain for the wild-type receptor and with introduction of the GP mutation at position 512–513 and 513–514 with local disruption of the helix (**b**). STAT5 transcriptional activity in the absence or presence of thrombopoietin (Tpo) of the indicated mTpoR (**c, d**) or hTpoR mutants (**e**) in γ–2A cells. Shown are averages of three independent experiments each done with 2–3 biological replicates ± SEM, Kruskal–Wallis nonparametric test with multiple-comparisons Steel's test with control (jmp pro12); *p<0.05, **p<0.01. Short-term proliferation assay (Cell Titer Glo, Promega) was performed on stable Ba/F3 cell lines expressing the indicated mTpoR mutants stimulated or not with Tpo (0.1, 1, 10, and 50 ng/ml) or IL-3 (20 ng/ml) (**f, g**). Values shown represent the average of three, respectively two, independent experiments each done with three biological replicates ± SEM., Kruskal–Wallis nonparametric test with Dunn's multiple-comparisons test (Prism 9.1.2); **p<0.01, ***p<0.001, ****p<0.0001, ns: nonsignificant.

The online version of this article includes the following source data and figure supplement(s) for figure 4:

**Source data 1.** Raw data, scatter plot, and statistics (Prism 9.1.2, jmp pro12) for *Figure 4c*.

**Source data 2.** Raw data, scatter plot, and statistics (Prism 9.1.2, jmp pro12) for *Figure 4d*.

**Source data 3.** Raw data, scatter plot, and statistics (Prism 9.1.2, jmp pro12) for *Figure 4e*.

**Source data 4.** Raw data, scatter plot, and statistics (Prism 9.1.2) for *Figure 4f*.

**Source data 5.** Raw data, scatter plot, and statistics (Prism 9.1.2) for *Figure 4g*.

**Figure supplement 1.** W491A mutation inhibits activation by Gly-Pro substitutions after the RWQFP motif.

**Figure supplement 1—source data 1.** Raw data, scatter plot, and statistics (Prism 9.1.2, jmp pro12) for *Figure 4—figure supplement 1*.

The results depicted in *Figure 4* indicate that disruption of helical structure by Gly-Pro substitutions is tolerated at the two positions immediately downstream of W515, and furthermore that these substitutions induce constitutive activation up to position 514 in mTpoR (521 in hTpoR) (*Figure 4c and e*). Indeed, hTpoR with a helical break at position 519/520 (RWQFP**GP**) constitutively activates the receptor (*Figure 4e*). Likewise, a helical break at position 512/513 (KWQFP**GP**) in the mTpoR induces activation of the receptor as confirmed in BaF3 proliferation experiments (*Figure 4f and g*). Past this position, breaking the helix at residues 515/516±518/519 (Y**GP**LRH and Y**GP**L**GP**) impairs the normal function of mTpoR (*Figure 4c*). We also found that substitution of these same residues to alanine (Ala) is well tolerated, consistent with the need for helical structure and not for the particular residues in that region (*Figure 4d*). Thus, as predicted by the NMR data on mutant TpoRs carrying upstream oncogenic mutations, loss of α-helicity in the region around and immediately past W515 (i.e. until residue 521, hTpoR nomenclature) can induce TpoR autoactivation. The loss of helical structure must, however, be localized as loss of α-helicity in the region starting six residues downstream of W515 profoundly impairs the function of hTpoR.

Next, we asked whether activation induced by these Gly-Pro mutants shares the same requirements as the canonical S505N and W515K hTpoR mutations. Those were shown to absolutely require the presence of Trp at position 491 in the extracellular JM region (*Levy et al., 2020*). In *Figure 4—figure supplement 1*, we show that the W491A mutation inhibits constitutive activation of the hTpoR RWQFPGP mutant, as it does for S505N (*Levy et al., 2020*). Thus, like the familial TM-JM mutations, the GP active mutant requires W491.

Together the results in *Figures 1 and 2* indicate that both the rotational orientation and the membrane proximity influence the activity of the Asn mutants. *Figures 3 and 4* suggest that the closer the mutation-induced dimerization of the TM helix is to the cytosolic border, the more pronounced is the local loss of α-helicity of the downstream helix that induces receptor activation.

## Alanine insertions in the context of the Asn-mutated receptors show that active mutants can be inactivated by rotation

The influence of rotation can be tested with Ala insertions before the KWQFP motif (*Figure 5a–c*). Although these substitutions both increase the helix length and change the rotational orientation, the greater influence is likely on helix rotation, which, as confirmed with AlphaFold2 (*Jumper et al., 2021*; *Varadi et al., 2022*), is predicted to change by 206° clockwise with +2 Ala insertions. When the insertion was added to the wild-type mTpoR, we detected a weak but significant increase in STAT5 transcriptional activity (*Figure 5d*). This is predicted by turning the receptor from the inactive WT conformation to one of the active conformations (*Figure 5c*) as previously defined by coiled coil (cc) fusion proteins (*Staerk et al., 2011*). These cc-mTpoR fusion constructs described in *Staerk et al., 2011* were engineered using the Put3 coiled coil dimerization domain fused in different registers with

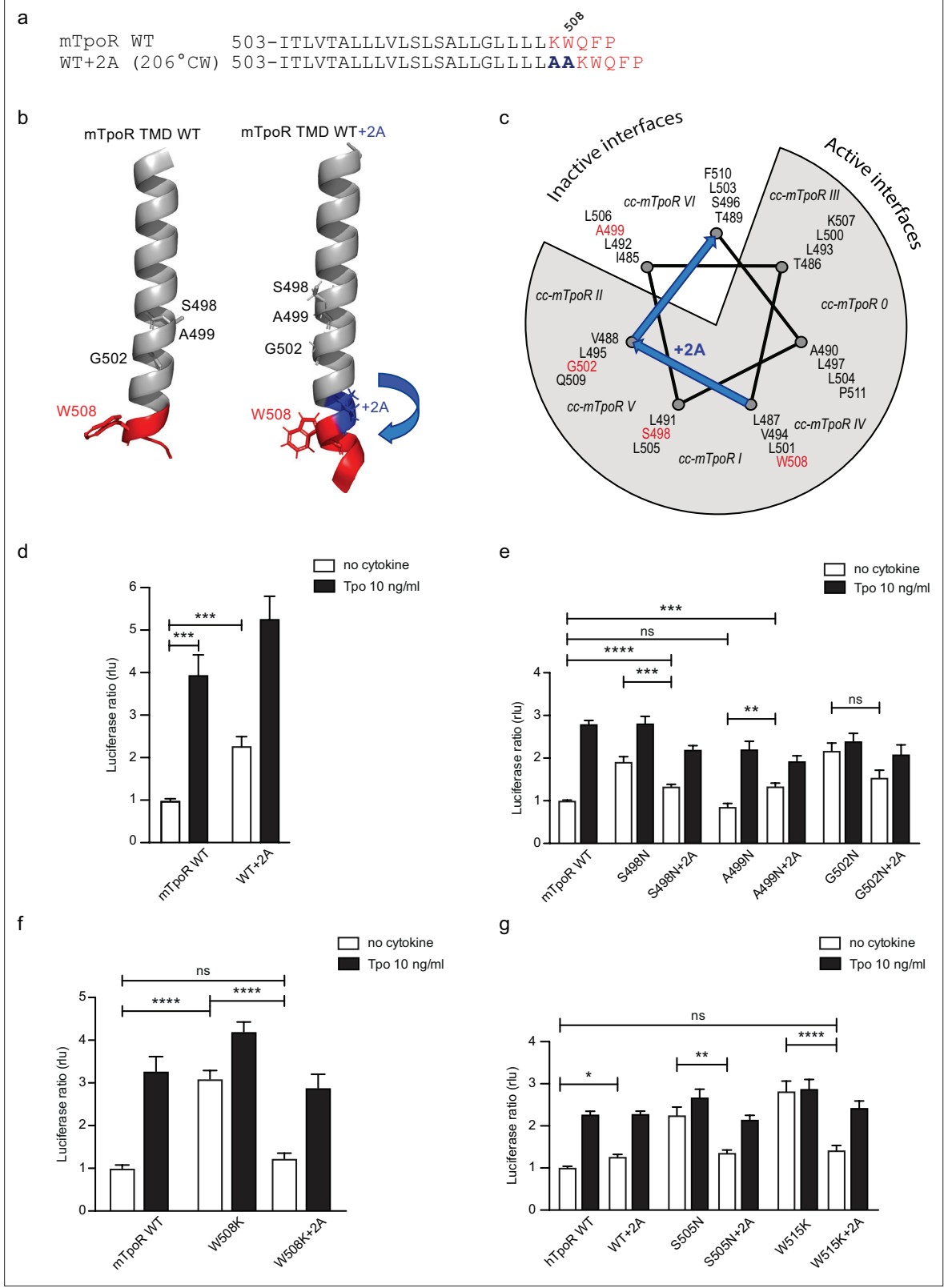

**Figure 5.** Effects of +2 alanine insertions on asparagine (Asn) and W508K mutants. Alanine (Ala) insertion mutagenesis in the murine thrombopoietin receptor (mTpoR) at the indicated position is predicted to extend the helix and rotate clockwise by 206° (**a**). Model generated with AlphaFold2 (*Jumper et al., 2021*; *Varadi et al., 2022*) of wild type (WT) mTpoR transmembrane domain (TMD) (right panel) and with insertion of 2 Ala (right panel) (**b**). Helical wheel diagram showing the positions of residues I485 to F511 relative to active and inactive interfaces of mTpoR. In italic are indicated the

*Figure 5 continued on next page*

*Figure 5 continued*

positions of coiled coil (cc) fusion TpoR constructs (cc-mTpoR) each adopting a particular transmembrane dimeric interface as described in *Staerk et al., 2011* (**c**). Spi-Luc STAT5 transcriptional activity in the absence or presence of thrombopoietin (Tpo) for the indicated mTpoR (**d–f**) or human TpoR (hTpoR) (**g**) WT or mutant constructs. Shown are averages of 3–10 independent experiments, each done with three biological replicates ± SEM in HEK293T cells and JAK2-deficient γ–2A cells; Kruskal–Wallis nonparametric test with multiple-comparisons Steel's test with control (jmp pro12); *p<0.05, ***p<0.001, ns, nonsignificant.

The online version of this article includes the following source data for figure 5:

**Source data 1.** Raw data, scatter plot, and statistics (Prism 9.1.2, jmp pro12) for *Figure 5d*.

**Source data 2.** Raw data, scatter plot, and statistics (Prism 9.1.2, jmp pro12) for *Figure 5e*.

**Source data 3.** Raw data, scatter plot, and statistics (Prism 9.1.2, jmp pro12) for *Figure 5f*.

**Source data 4.** Raw data, scatter plot, and statistics (Prism 9.1.2, jmp pro12) for *Figure 5g*.

the mTpoR TM-cytosolic domains, which provided a way to control dimerization via a specific TM domain interface.

We added the +2 Ala insertion to the inactive Asn receptor mutant (A499N) and to the two active receptors mutants (S498N and G502N) (*Figure 5e*). The +2 Ala insertion is supposed to rotate N499 from the inactive orientation toward an active dimeric interface. In contrast, the +2 Ala insertion is supposed to rotate N498 and N502 toward interfaces shown to be slightly less active, but not to a completely inactive interface (*Staerk et al., 2011*). Remarkably, with the +2 Ala insertion we detected, using the STAT5 transcriptional activity assay, a significant increase in activity of the A499N mutant, a small decrease in activity of the S498N, and much weaker effect on the G502N mutant, as predicted (*Figure 5e*; *Staerk et al., 2011*).

In addition to these studies, we also measured the $^{13}$C=O chemical shift of L505 in the A499N mTpoR construct with and without the +2 Ala insertion (*Figure 3i and j*, respectively). The L505 $^{13}$C=O chemical shift was at 176.4 ppm in the A499N construct characteristic of helical secondary structure, but decreased to 174 ppm with the +2 Ala insertion, indicative of localized loss of helical structure, in agreement with activation.

## The W508K mutant can be inactivated by rotation into a single inactive conformation

We have previously described how mutations of W508 (W515 in hTpoR) can activate the receptor (*Pecquet et al., 2010*; *Staerk et al., 2006*). W508 is expected to be located at the membrane boundary and its mutation may be related to the observation that the Asn mutations closer to the membrane boundary have higher activity. To uncouple the effects of rotational orientation and proximity to the membrane surface, we assessed the effect of +2 Ala insertions in the context of the W508K mutation. Strikingly, the +2 Ala insertion exerted strong inhibition on W508K (*Figure 5f*), which perfectly fits the orientations determined by cc-TpoR fusions, as the +2 Ala insertion will rotate the W508/515 mutants to the inactive interface (*Figure 5c*). The +2 Ala insertion inhibited the W508K mTpoR mutants to a higher extent than the S498N and G502N mutants, according to the interface induced (*Figure 5c*). Importantly, the +2 Ala insertion inhibited hTpoR W515K as well (*Figure 5g*). Last but not least, we measured the $^{13}$C=O chemical shift of L512 in the W515K construct of hTpoR sequence with the +2 Ala insertion and detected a downfield chemical shift to 175.6 ppm, indicating that the +2 Ala insertion led to acquisition of helical structure like in the non-mutated TpoR (*Figure 3g*). In line with the results in mTpoR, the insertion of +2 Ala led to a small but significant decrease in signaling by the constitutively active S505N mutant (*Figure 5g*). Thus, rotation of the JM domain by insertion of +2 Ala activates the inactive TpoR A499N, while it decreases signaling by mTpoR S498N and hTpoR S505N, and inhibits the W508/W515 mutants.

## TpoR transmembrane domain dimerization induces membrane binding of the C-terminal switch region

TpoR, like other cytokine receptors, binds the JAK2 FERM domain via the region between Box 1 and Box 2 in the cytosolic domain (*Huang et al., 2001*; *Royer et al., 2005*). However, the control over JAK2 activation is exerted via the first 11–15 JM cytosolic residues, of which the switch motif I$^{528}$W$^{529}$ upstream of Box 1 is crucial (*Constantinescu et al., 2001a*). Recently, it has been shown that the

equivalent of W529 in hTpoR from EpoR and leptin receptor interacts with the pleckstrin-like subdomain F3 of the FERM domain of JAK2 (with and around W298) in a trans-configuration explaining how these residues do not contribute to the receptor-JAK2 affinity but rather regulate activation as they bridge two JAK2s in an active dimer (*Ferrao et al., 2018*). Were this model to be valid for TpoR, we would predict that in the inactive state of TpoR WT, this switch residue W529 would be accessible in solution, but upon activation in hTpoR S505N or W515K mutants, this W529 residue would make contact to the membrane or the FERM domain of the other JAK2 molecule, which is very close to the cytosolic leaflet of the plasma membrane. A recent study showed that indeed FERM domain residues interact with the membrane inner leaflet (*Wilmes et al., 2020*).

The studies undertaken above show that both the S505N and W515K mutations lead to dimerization in active orientations, although the dimer structures are likely not identical. Membrane interactions between the JM sequences rich in basic and hydrophobic residues are often observed in single pass membrane proteins (*Zhang et al., 2006*). The INEPT (Insensitive Nuclei Enhancement by Polarization Transfer) NMR experiment can be used to address membrane binding in the JM region of the TM-JM peptides (*Matsushita et al., 2013*). For example, we have previously shown that the JM region of the Neu receptor tyrosine kinase binds to the membrane in the inactive receptor dimer and releases from the membrane upon receptor activation (*Matsushita et al., 2013*).

*Figure 6* presents the results from a similar INEPT NMR experiment in the hTpoR. In this experiment, polarization is transferred from $^1$H to $^{13}$C through J-couplings, and $^{13}$C NMR resonances are enhanced at protein sites that are mobile, having correlation times of <0.01 µs. The $^{13}$C INEPT spectra in *Figure 6* were obtained at 37°C using hTpoR TM-JM peptides reconstituted into phospholipid bilayers comprised of dimyristoylphosphocholine and dimyristoylphosphoglycine in a 10:3 molar ratio. The hTpoR peptides contain a single uniformly -$^{13}$C labeled amino acid (W529) at the C-terminus of the switch region that serves as a probe for membrane binding. The $^{13}$C resonances between 10 and 80 ppm correspond to natural abundance $^{13}$C sites in the lipids. At 37°C, these lipids are highly dynamic. The aromatic $^{13}$C resonances of U-$^{13}$C-labeled W529 are also observed between ~110–130 ppm, suggesting that in the WT TpoR the W529 side chain is mobile and does not interact with the membrane surface. In contrast, in both the S505N (*Figure 6b*) and W515K (*Figure 6c*) TM-JM peptides, the $^{13}$C resonances from W529 are lost, consistent with membrane binding.

## Influence of TpoR activation on the proximity between JAK2 kinase domains

Our model based on the above results is that localized loss of α-helicity ultimately allows activation of the kinase domain of JAK2 (named JH1), namely trans-phosphorylation and activation. We next addressed how receptor unraveling is correlated with JAK2 activation and with the JH1 domains getting into close proximity. The question is whether the increased activity observed with Asn substitutions closer to the membrane surface is correlated with close apposition of the JAK2 JH1 domains. To answer this question, we employed a protein to protein interaction (PPI) assay based on *Gaussia princeps* split-luciferase reversible complementation (*Remy and Michnick, 2006*).

We introduced either of the two complementing fragments of *Gaussia* luciferase (Gluc1 and Gluc2) (*Remy and Michnick, 2006*) at the C-terminus of the JH1 domain of separate full-length JAK2 constructs (*Figure 6d*). This configuration allows interpretation of the luminescence measurements as a quantification of the dimerization of JAK2 proteins via their C-terminal JH1 kinase domains.

The JAK2 fusion proteins were co-expressed with mTpoR WT or with G502N (*Figure 6e*). The latter induces a significant increase in JAK2 JH1 dimerization. We next tested JAK2 JH1 dimerization with the mTpoR W508K or mTpoR A499N mutants in the presence or absence of the +2 Ala insertion (*Figure 6f*). A499N is an inactive dimer as shown previously (*Leroy et al., 2016*). It is rendered active by the +2 Ala insertion as already discussed in *Figure 5e*. In contrast, the W508K mutation in mTpoR leads to higher dimerization of JAK2 JH1 domains, which is reversed by the +2 Ala insertion (*Figure 6f*). Furthermore, dimerization of JAK2 was not dependent on the kinase activity of JAK2 but on the actual physical proximity of the JH1 domains, as shown in *Figure 6—figure supplement 1* using the kinase dead K882D version of JAK2. That the mTpoR W508K mutant induces higher dimerization of the JAK2 JH1 domains compared to that induced by wild-type mTpoR (*Figure 6f*) was expected, knowing that the murine W508K mutant corresponds to the W515K mutant of hTpoR, which induces ligand-independent receptor dimerization and constitutive STAT5 activation by the luciferase reporter

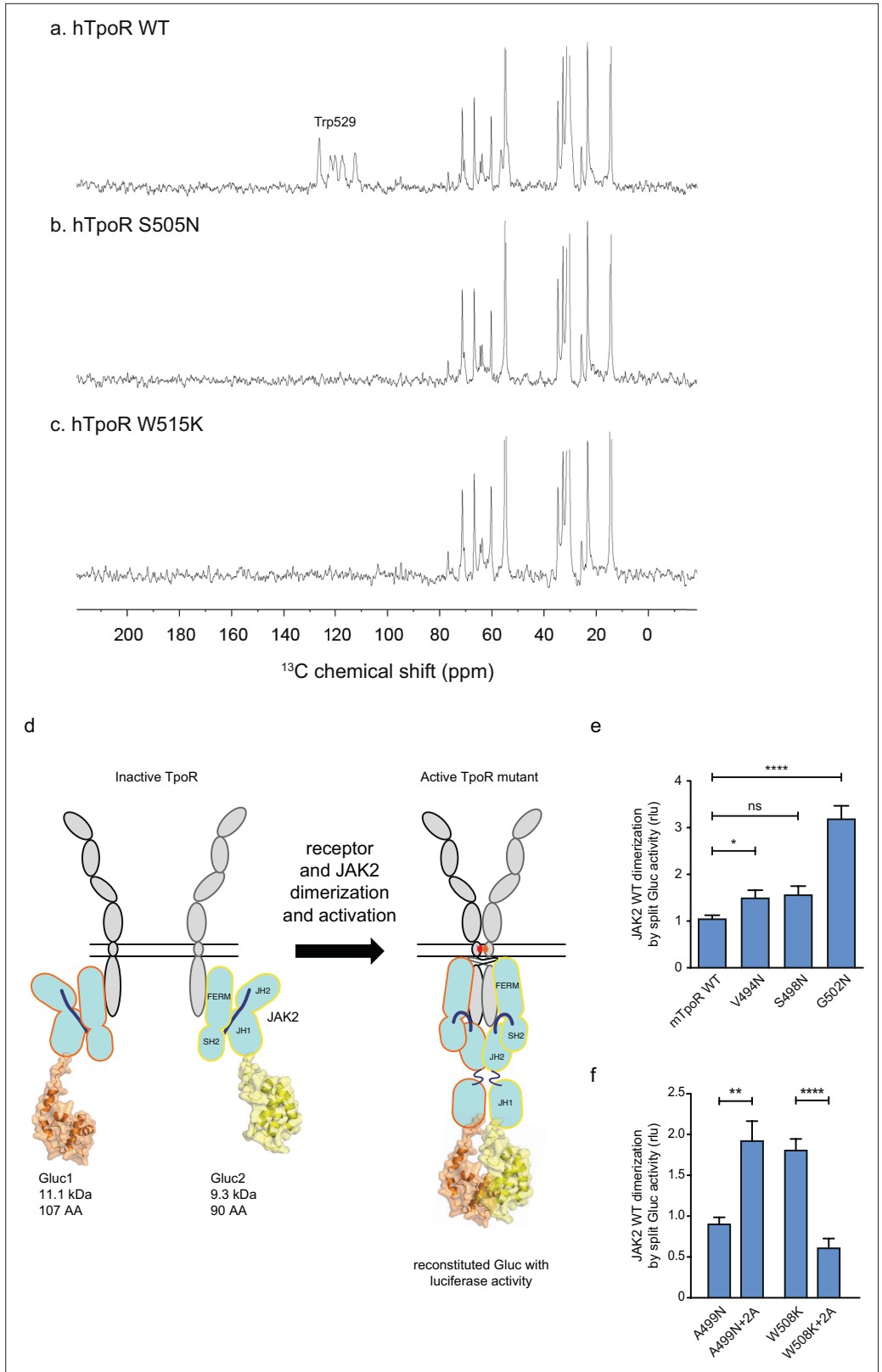

**Figure 6.** Thrombopoietin receptor (TpoR) transmembrane domain dimerization induces membrane binding of the C-terminal switch region and influence the proximity between JAK2 domains. Solid-state $^{13}$C NMR experiments using magic angle spinning and the INEPT pulse sequence were used to address membrane binding in the juxtamembrane (JM) region of the transmembrane and juxtamembrane (TM-JM) peptides of human

*Figure 6 continued on next page*

*Figure 6 continued*

TpoR (hTpoR). INEPT spectra were obtained at 37°C using hTpoR reconstituted into DMPC:PG bilayers. TpoR peptides were U-$^{13}$C labeled at W529 at the C-terminus of the switch region. Two independent technical replicates (reconstitutions and NMR data sets) were obtained for each experiment (**a–c**). Dimerization of JAK2 bound to the indicated murine TpoRs (mTpoR) was assessed by *Gaussia* luciferase in HEK-293T cells (**d–f**). Shown are averages of separate experiments ± SEM (n = 8); each experiment being performed with three biological repeats for each condition (triplicates). Kruskal–Wallis nonparametric test with multiple-comparisons Steel's test with controls (jmp pro12); **p<0.01, ****p<0.0001.

The online version of this article includes the following source data and figure supplement(s) for figure 6:

**Source data 1.** Raw data, scatter plot, and statistics (Prism 9.1.2, jmp pro12) for *Figure 6e*.

**Source data 2.** Raw data, scatter plot, and statistics (Prism 9.1.2, jmp pro12) for *Figure 6f*.

**Source data 3.** Raw data, scatter plot, and statistics (Prism 9.1.2, jmp pro12) for *Figure 6f*.

**Figure supplement 1.** Dimerization of JAK2 is influenced by murine thrombopoietin receptor (TpoR) transmembrane mutations irrespectively of JAK2 activity.

assay (*Figure 5g*). Following this observation, we also show that the inactive W508K+2 Ala mutant induces a lower dimerization of JAK2 JH1 (*Figure 6f*).

In conclusion, the dimerization signals of JAK2 kinase domains fit well with the functional data in STAT5 transcriptional assays. Of note, in these assays, the rotation induced by the +2 Ala insertion activates the A499N mutant and inactivates the W508K mutant, indicating that such rotations directly impact the next downstream signaling event, which is represented by dimerization and activation of JAK2.

## A consensus mechanism of activation by TM and JM mutations involves local loosening of restrictions at the intracellular boundary

The combination of the Ala insertion data and NMR data presented here suggests that the change in rotation and close dimerization of TM helices translate into a local loss of helical structure at the intracellular TM boundary. We propose that this provides greater flexibility in allowing the switch residues of TpoR, especially W529, to contact the JAK2 on the other receptor chain in the dimer and eventually to reorient the bound JAK2 molecules (*Figure 7*). This mechanism appears to be required for activating mutations that either substitute Asn for certain TM residues or remove Trp from the key 515 position.

For the human GHR, which does not contain the unique R/KWQFP insert, it has been shown that upon ligand binding activation promoted by dimerization of TM helices leads to separation of distal cytosolic receptor sequences and release of restrictions on the proximity of bound JAK2 (and possibly its relative orientation) (*Brooks et al., 2014*). Activation of GHR was associated by transition from parallel TM domains to a crossed structure of TM domains with increased tilt. The structural basis for such a mechanism is not known. For TpoR, we and others have observed that activation decreases the tilt of TM helices, with active mutants exhibiting more parallel orientations (*Leroy et al., 2016*). Thus, the GHR and TpoR adopt opposing TM tilt characteristics. Here we show that several mutations leading to activation of TpoR induce partial loss of the α-helix around the JM K/RWQFP motif, and this is proportional to the proximity to the cytosolic border of the TM Asn substitutions. Thus, for TpoR, the mechanism to achieve separation near the start of the cytosolic domain is a loss of the α-helix, unlike for GHR which adopts a crossed TM dimer structure in an active state. In agreement with *Ferrao et al., 2018*, both mechanisms would result in sufficient flexibility to allow binding of receptor switch residues to the F3 domain of the other JAK2 in an active complex (*Figure 7*).

To explore the role of TM rotations, we introduced extra Ala residues before the KWQFP motif in the JM region of the mTpoR. For these mutants, we assessed dimerization of JAK2 fragments and the secondary structure around W508. We show that rotation induced by +2 Ala insertions can inhibit constitutive activity of the Asn and W508K mutants. Using peptides corresponding to the TM-JM regions of TpoR, we show here that oncogenic mutants S498N and G502N in mouse, and W515K in human (*Leroy et al., 2016*), exhibit local loss of α-helicity around the K/RWQFP motif (*Figure 3*). Interestingly, the +2 Ala insertion also exhibits a partial loss of secondary structure around the K/RWQFP motif compared to the wild type sequence, possibly explaining the weakly activating phenotype

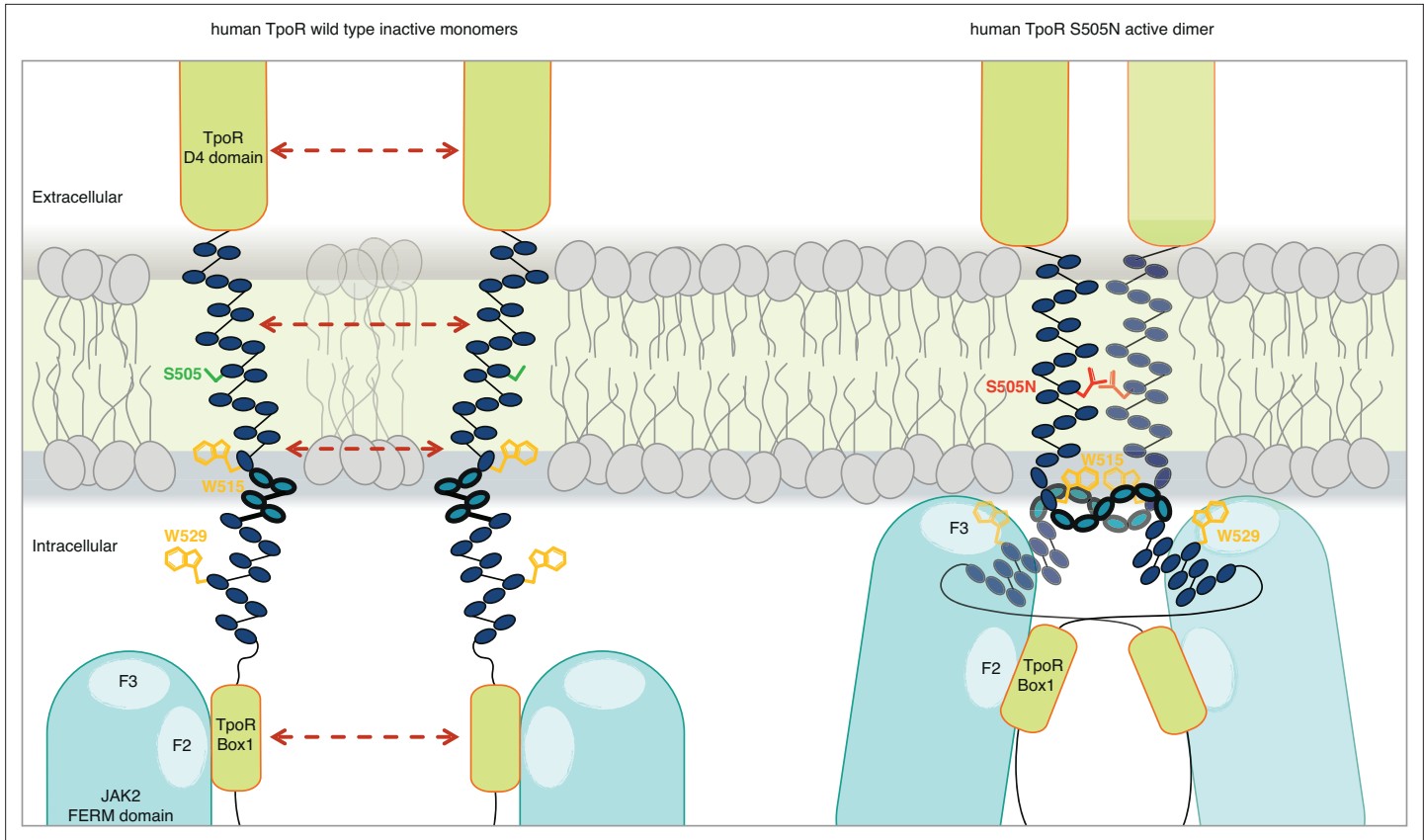

**Figure 7.** Canonical mutations S505N and W515K of the human thrombopoietin receptor (TpoR) found in myeloproliferative neoplasms activate the receptor in absence of its ligand by inducing a localized loss of helicity in the intracellular juxtamembrane domain. This provides greater flexibility to allow the switch residues of TpoR to contact the F3 subdomain of the JAK2 appended to the other receptor chain in the dimer and eventually reorient the bound JAK2 molecules. Cartoon depicting human TpoR wild type inactive monomers and canonical S505N mutant in an active dimer conformation, with the appended JAK2 molecules.

of +2 Ala insertion on TpoR (*Figure 5d*). Reversal of activation of hTpoR W515K by the +2 Ala insertion correlates well with restoration of α-helical structure in this double mutant around W515 (*Figure 3*).

Noteworthy, the A499N mutant, which adopts an inactive dimeric configuration, can be activated by the rotation imposed by the +2 Ala insertion, supporting the notion that activation by TM and JM mutations requires dimerization and rotation.

Our data show that proximity and orientation of the TM domains are both required for activation of the TpoR. Single amino acid substitutions in the TM domains with Asn induce an increasingly important loss of α-helicity in the JM helix as these substitutions edge closer to the cytosolic domain of the receptor. The unraveling of the JM helices coincides with activation, but only if the register of the helices where the Asn substitutions are introduced matches the steric rotational constraints for proper assembly of the JH1 kinase domains of the JAK2 molecules that are bound downstream of the receptor chain. Thus, activation is controlled by liberation of JAK2 constraints by partial unraveling of JM helices, leading to a precise reconfiguration of JAK2 domains and allowing phosphorylation of the activation loop Y1007 by the opposite JAK2.

As JAK proteins are composed of four different domains, one can assume that the different domains of JAK2 can fulfill different functions regarding the receptor/JAK2-complex activation. It is not known whether only the FERM domain or also either the pseudokinase, the kinase, or the SH2 domain of JAK2 might be linked to receptor unraveling. Hydrophobic sequences of the 'switch' motif at the outset of Box 1 in TpoR (*Constantinescu et al., 2001a*) would be good candidates for interaction with the kinase or pseudokinase domain in an inactive receptor/JAK complex and possibly interaction with the FERM domain in the active configuration. Recently, using X-ray crystallography on segments of JAK2 and cytosolic domains of cytokine receptors, it was shown that the W

residues of the switch motifs in the erythropoietin and leptin receptors may interact with the FERM domain of the JAK2 appended to the other receptor monomer, thus presumably favoring activation via formation of a cross-structure where one receptor binds two JAKs, one via a constitutive F2 binding and the other via the activation-induced binding of switch residues to F3, where F2 and F3 refer to subdomains of the FERM domain, respectively, an acyl CoA-binding protein-like domain (F2) and a Pleckstrin Homology (PH)-like fold (F3) (*Ferrao et al., 2018*). In this model, the switch residues have an important contribution to bridging the receptor-JAK2 into active dimers (*Ferrao et al., 2018*), in agreement with the original model we proposed for switch residues, which act in a distinct way from the residues that recruit JAK2 to the receptor (*Huang et al., 2001*; *Constantinescu et al., 2001a*). While Box 1 was defined as proline-rich sequences (PXXPXP) that may include the hydrophobic switch residues, recent structural work argue that while the PXXPXP motif binds F2 (*Wallweber et al., 2014*; *Zhang et al., 2016*), the upstream switch residues bind F3 (*Ferrao et al., 2018*), thus targeting distinct regions of JAK2 and suggesting that they fulfill different functions as distinct motifs.

Loss of the α-helix between the TpoR TM domain and Box 1 would allow this cross configuration with the switch residue W529 interacting in the membrane with the F3 FERM domain of the JAK2 in trans. Results of our INEPT NMR experiment agree with a loss of rigidity upon activation. Localized loss of α-helicity in the cytosolic JM domain and significant conformational change around switch residue W529 are likely to trigger the liberation of the kinase domains of JAK2 from physical sequestration by the pseudokinase and FERM domains. More experiments are needed to determine whether this process is asymmetric with respect to which receptor and which JAK2 initiate activation, depending on which receptor subunit unravels first. Also, more investigation is required before concluding that such unraveling may occur for receptors such as EpoR or TpoR in complexes with the myeloproliferative neoplasm-prevalent JAK2 V617F mutant, for which a dimeric model structure was recently obtained (*Glassman et al., 2022*).

There are precedents for secondary structure changes of cytosolic JM regions in membrane receptors. A mutated Neu (HER2) receptor tyrosine kinase in breast cancer was shown to exhibit a change from random coil to helix when compared to WT HER2, leading to activation of the intrinsic kinase domain (*Matsushita et al., 2013*). Random coil structure in the JM region was found to bind the lipid headgroups of the inner leaflet of the membrane and hold the receptor in an inactive conformation. Also, the bacterial DesK TM receptor contains in its cytosolic region a catalytic domain that functions as a phosphatase or a kinase according to the temperature. Kinase function was associated with helical structure and phosphatase function with random coil structure induced by interaction with the lipids at the inner leaflet of the membrane (*Inda et al., 2014*). The sequence of DesK is charged and contains both positive and negative charges, the former are required for binding to the negative charges of lipids. Mutation of positive charges to Ala results in constitutive helical structure and kinase activity at low temperature (*Inda et al., 2014*). In contrast, our results show that the cytosolic JM domain of a cytokine receptor switches from helical in the inactive state into random coil and membrane lipid bound in the active state. This switch is associated to interaction with lipids and is likely to liberate JAK2 proteins from a relative position that is inactive into an active position. Our model is compatible to the model of *Brooks et al., 2014* where flexibility of the cytosolic domain leads to JAK2 activation, while we show that this is localized, and we delineate the borders of this region that loses helicity. Our results are also compatible with the model proposed by *Wilmes et al., 2020* where monomeric TpoR needs to be dimerized for activation. This model fits with our recent results showing that JAK2 V617F can induce close apposition of the cytosolic domains of cytokine receptors, such as EpoR (*Leroy et al., 2019*).

While our previous data and this study suggest that the small molecule agonist eltrombopag used in the clinics to activate TpoR also dimerizes the TM domains (*Leroy et al., 2016*) and leads to unraveling of the cytosolic JM region, an important question is whether this local loss of α-helicity around the JM RWQFP motif is also required for the physiological activation by cytokine. Tpo is predicted to impose a massive re-orientation of the extracellular domains, and it is possible that a dimeric configuration leading to JAK2 activation can be adopted without close contacts between TM domains. Of note, Tpo is able to weakly activate the mTpoR A499N inactive pre-formed dimer. This observation suggests that extracellular domain re-configuration can overwhelm the stabilizing interactions between TM domains.

Since the +2 Ala insertion rotating the JM helix by +206° inhibits pathogenic W515K mutants, our data open the way for the possibility that extracellular proteins such as antibodies, nanobodies, or diabodies could re-configure the pathological receptor dimerization and inhibit pathological activation by Asn or W515X mutations in myeloproliferative neoplasms. Recently, diabodies targeting the extracellular domain of EpoR were identified that maintain the receptor monomers at distances compatible with weak activation of WT JAK2 but induce inactivation of JAK2 V617F-bound EpoR (*Moraga et al., 2015*). A similar functional effect was recently described for diabodies targeting the extracellular domain of TpoR; however, no structural data exist for the precise dimeric interface(s) (*Cui et al., 2021*). These data suggest nevertheless that reconfiguring the extracellular domain of cytokine receptor mutants might have therapeutic significance.

Finally, our data build on the rich literature regarding the role of tryptophan at the membrane–water interface. Tryptophan has two unique properties that form the basis of its structural and functional roles in proteins. First, the side chain has both strong hydrophobic and hydrophilic character as a result of the aromatic rings and NH functional group of its indole side chain. In membrane proteins, the side chain is localized to the head-group region of the bilayer (*Yau et al., 1998*). This unique character of the side chain is elegantly exhibited in studies by *Braun and von Heijne, 1999*, who found that Trp, but not Phe, pulls model TM helices toward the lipid–water interface. The strong partial negative charge resulting from the aromatic ring current on the face of the indole side chain is often found associated with basic amino acids. These cation-$\pi$ interactions are an essential for partitioning of positively charged residues into the membrane bilayer in antimicrobial peptides. The interaction of R414 and W515 in the RWQFP motif allows this region to adopt helical secondary structure at the end of the TpoR TM helix. The second unique property of the tryptophan side chain is its large size, which is not easily tolerated in the well-packed interior of proteins. When found in the interior of membrane proteins, tryptophan is stabilized by a number of specific interactions that reflect its dual hydrophobic–hydrophilic character. For example, a highly conserved tryptophan plays a unique role in G protein-coupled receptors for controlling helix unraveling that is associated with receptor activation (*Pope et al., 2020*). In the TpoR, activation results in the rotation of Trp515 into the helical interface of the TM dimer, where it is not tolerated due to steric interactions. Together these studies provide insights into why Trp is unique within the RWQFP motif and why mutations to other amino acids lead to constitutive receptor activity.

## Materials and methods

### Plasmid constructs

The murine and human TpoR wild types (WT), the Asn substitution, and Ala insertion mutant constructs were subcloned into pMX-IRES-GFP as described previously (*Staerk et al., 2006*). All the TpoR constructs contained a hemagglutinin Tag (HA-Tag) at the N-terminus downstream of the signal peptidase cleavage site (*Huang et al., 2001*). Mutagenesis was achieved by overlapping extension PCR using the Pfu Turbo DNA polymerase (Stratagene, La Jolla, CA). For the split luciferase experiments, cDNAs coding for the amino acids 1–93 (hGluc1) or the amino acids 94–169 (hGluc2) of *Gaussia princeps* luciferase (*Remy and Michnick, 2006*) were inserted downstream of the cDNA encoding JAK2, following cDNA coding for a (GGGGS)$_2$ flexible linker. These constructs were subcloned into the pcDNA3.1/Zeo vector. All constructs were verified by Sanger sequencing (Macrogen, NL). Sequences of plasmids and primers are accessible in *Supplementary file 1*.

### Dual luciferase transcriptional assays

Transcriptional activity of STAT5 was analyzed in HEK293T cells and JAK2-deficient γ–2A cells (*Supplementary file 2*) co-transfected with the receptors variants, JAK2 WT, STAT5b, and Spi-Luc or pGRR5 (*Sliva et al., 1994*). The pRL-TK vector (Promega, Madison, WI) was used as an internal control. Luminescence was measured in cell lysates 24 hr after transfection using the Dual Luciferase Reporter Assay Kit (Promega) on a Victor X luminometer (PerkinElmer) or a GloMax Discover Multimode Reader.

### Split *Gaussia* dual-luciferase assays

HEK-293T cells were plated in a 24-well plate at 400,000 cells per well before being transiently transfected with pGL3-control (Promega), a construct constitutively expressing a firefly luciferase, and

with two constructs expressing either Gluc1 or Gluc2 split *Gaussia* luciferase subunits fused to JAK2 (*Remy and Michnick, 2006*). Transfection was performed using the Transit-LT1 transfection reagent from Mirus Bio (distributed by Sopachem, Eke, Belgium). Cells were grown in DMEM + 10% FBS for 48 hr post-transfection before lysis. Lysis was carried out by replacing media with 120 µL/well of F12/DMEM (no phenol red) +1X protease inhibitor cocktail (Halt from Thermo Fisher, Aalst, Belgium) before performing two freeze/thaw cycles at –80°C and 37°C, respectively. Lysates were centrifuged at 13,200 RPM on a table-top centrifuge for 5 min, and then 50 µL of supernatant was transferred to a white opaque 96-well plate. Luciferase readings were performed on a Victor X luminometer. Using the reagents from the Dual Luciferase Reporter Assay Kit, first 35 µL of LAR II reagent was added per well to read the firefly luciferase signal (transfection control). Next, 35 µL of the Stop & Glo reagent (which contains *Gaussia* luciferase substrate coelenterazine) was added per well to stop the firefly luciferase activity and read the reconstituted *Gaussia* luciferase signal. The final readout was calculated as *Gaussia* luc signal (RLU)/Firefly luc signal (RLU) × 1000.

## Generation of cell lines

Ba/F3 were transduced with the TpoR constructs using BOSC23 ecotropic viral supernatant (*Liu et al., 2000*). Cell populations expressing GFP at a level above 15% were selected by flow cytometry. Cells were grown in IL3-containing RPMI supplemented with 10% FBS.

## Cell viability assays

The proliferation of Ba/F3 expressing WT or mutants TpoR mutants was assessed by Cell-Titer-Glo luminescent cell viability assay (Promega). Prior to the experiment, cells were washed three times in PBS. 96-wells plate were used to plate the cells at 10,000 cells/well in RPMI +10% FBS supplemented or not by different concentrations of recombinant Tpo (R&D Systems). After 72 hr, ATP-reacting substrate was added following the manufacturer's instructions and luminescence readings were performed using a Perkin Elmer Victor X luminometer.

## Bone marrow reconstitution

Male C57BL/6 mice bone marrow was isolated and cultivated in the presence of cytokines (SCF, Flt-3, Tpo, IL-3, and IL-6). These cells were spin infected with 300 µL concentrated VSV-G pseudo-typed retroviral supernatant expressing mTpoR WT or mutants, obtained as previously described in *Pecquet et al., 2010*. $10^6$ infected cells were then injected intravenously in lethally irradiated female mice aged between 6 and 8 weeks old. After bleeding, red blood cell (RBC), platelets, white blood cells (WBC), and neutrophils were measured using an MS9 blood cell counter at day 40 post-transplantation. GFP was checked at multiple time points to evaluate chimerism. For histological and morphological analysis, mice were sacrificed after bleeding at day 40 post-transplantation. Slides were analyzed with CaseViewer 2.4 (3DHISTECH Ltd.). This work was approved by the Ethics Committee for Animal Experimentation of the Université catholique de Louvain under the reference 2019/UCL/MD/026 (*Supplementary file 3*).

## Solid-state NMR spectroscopy

For solid-state NMR studies on TpoR peptides reconstituted into model membrane bilayers, 44 residue peptides corresponding to the TM domain and JM regions of TpoR were synthesized using solid-phase methods. The sequence for human TpoR was RRRETAWISLVTALHLVLGLSAVLGLLLLRWQ FPAHYRRLRHALWPS-NH$_2$, while the sequence for the murine TpoR was RRRETAWITLVTALLLVLSLSALL GLLLLKWQFPAHYRRLRHALWPS-NH$_2$. Both peptides contained C-terminal carboxy amide protecting groups. The peptides included three non-native Arg at the N-terminus for peptide solubility and to terminate both ends of peptide with positive charges (*Figure 3a*). The crude peptide (5–15 mg) was purified by reverse-phase HPLC on a C4 column using gradient elution and reconstituted by detergent dialysis using DMPC and DMPG (10:3) and an ~1:60 peptide to lipid ratio. Lipids were obtained from Avanti Polar Lipids (Alabaster, AL) as lyophilized powders and used without further purification. The DMPC, DMPG lipids with 14-carbon chains roughly match the hydrophobic thickness of typical cell membranes, while DMPG introduces a net negative charge. This lipid mixture has previously been used to demonstrate that the TM-JM peptides mimic the dimerization behavior of the full-length receptor (*Defour et al., 2013*; *Tang et al., 2019*), in which the wild-type human TpoR receptor is

monomeric while the W515K and S505N receptors are dimeric. Magic angle spinning NMR experiments were performed at a $^{13}C$ frequency of 125 MHz on a Bruker AVANCE spectrometer. The MAS spinning rate was set to 9–11 KHz (±5 Hz). The ramped amplitude cross-polarization contact time was 2 ms. Two-pulse phase-modulated decoupling was used during the evolution and acquisition periods with a radiofrequency field strength of 80 kHz. The sample temperature was maintained at 198 K (±2 K).

The INEPT pulse sequence was used to address membrane binding in the JM region of the TM-JM peptides of hTpoR as previously described (*Matsushita et al., 2013*). INEPT spectra were obtained at 37°C using hTpoR reconstituted into DMPC:PG bilayers. hTpoR peptides were U-$^{13}C$ labeled at W529 at the C-terminus of the switch region.

## Acknowledgements

TB was supported by a Télévie PhD fellowship. GL was supported by a PhD Fellowship awarded by the Fondation 'Les Avions de Sébastien.' SNC is Honorary Research Director at FRS-FNRS Belgium. Funding to SNC is acknowledged from the Ludwig Institute for Cancer Research, Fondation contre le cancer, Salus Sanguinis, Fondation 'Les Avions de Sébastien,' projects Action de recherche concertée (ARC) 16/21-073 and WelBio F 44/8/5-MCF/UIG-10955. NP has received an FSR PhD Fellowship from Université catholique de Louvain and an Aspirant PhD fellowship from the FRS-FNRS, Belgium. We gratefully acknowledge the WM Keck Foundation for support of the NMR facilities in the Center of Structural Biology at Stony Brook.

## Additional information

### Competing interests

Stefan N Constantinescu: is co-founder of MyeloPro Diagnostics and Research GmbH, Vienna, Austria. The other authors declare that no competing interests exist.

### Funding

| Funder | Grant reference number | Author |
|---|---|---|
| Fonds National de la Recherche Scientifique | Télévie PhD fellowship | Thomas Balligand |
| Fonds National de la Recherche Scientifique | | Nicolas Papadopoulos |
| Les avions de Sebastien | | Gabriel Levy |
| Ludwig Institute for Cancer Research | | Stefan N Constantinescu |
| Stichting Tegen Kanker | | Stefan N Constantinescu |
| Catholic University of Louvain | Salus Sanguinis | Stefan N Constantinescu |
| Les avions de Sébastien | | Stefan N Constantinescu |
| Action de recherche concertée | 16/21-073 | Stefan N Constantinescu |
| Walloon Excellence in Lifesciences and Biotechnology | F 44/8/5 - MCF/UIG - 10955 | Stefan N Constantinescu |
| W. M. Keck Foundation | | Steven O Smith |
| Stony Brook University | National Institutes of Health, RO1 GM 46732 | Steven O Smith |

The funders had no role in study design, data collection and interpretation, or the decision to submit the work for publication.

## Author contributions

Jean-Philippe Defour, Emilie Leroy, Conceptualization, Data curation, Validation, Investigation, Methodology, Writing – original draft; Sharmila Dass, Céline Mouton, Lidvine Genet, Investigation; Thomas Balligand, Gabriel Levy, Christian Pecquet, Conceptualization, Validation, Investigation, Writing – original draft; Ian C Brett, Data curation, Validation, Investigation; Nicolas Papadopoulos, Data curation, Software, Formal analysis, Validation, Investigation, Visualization, Writing – original draft, Writing – review and editing; Judith Staerk, Writing – review and editing; Steven O Smith, Supervision, Validation, Investigation, Methodology, Writing – original draft; Stefan N Constantinescu, Conceptualization, Resources, Supervision, Funding acquisition, Validation, Methodology, Writing – original draft

## Author ORCIDs

Emilie Leroy http://orcid.org/0000-0001-5897-1713
Gabriel Levy http://orcid.org/0000-0001-6746-3083
Nicolas Papadopoulos http://orcid.org/0000-0001-7869-862X
Steven O Smith http://orcid.org/0000-0003-1861-7159
Stefan N Constantinescu http://orcid.org/0000-0002-8599-2699

## Ethics

This work was approved by the Ethics Committee for Animal Experimentation of the Université catholique de Louvain under the reference 2019/UCL/MD/026 . For this specific work in the field of cancer research, pain and discomfort of the animals was monitored in strict accordance with the recommendations on best practice and commonly used reference in the field : Workman P, Aboagye EO, Balkwill F, Balmain A, Bruder G, Chaplin DJ, Double JA, Everitt J, Farningham DA, Glennie MJ, Kelland LR, Robinson V, Stratford IJ, Tozer GM, Watson S, Wedge SR, Eccles SA; Committee of the National Cancer Research Institute. Guidelines for the welfare and use of animals in cancer research. Br J Cancer. 2010 May 25;102(11):1555-77. doi: 10.1038/sj.bjc.6605642. PMID: 20502460; PMCID: PMC2883160.

## Decision letter and Author response

Decision letter https://doi.org/10.7554/eLife.81521.sa1
Author response https://doi.org/10.7554/eLife.81521.sa2

# Additional files

## Supplementary files

• Supplementary file 1. Plasmids and primers. List of plasmids used throughout the experiments and primers used for various mutagenesis.

• Supplementary file 2. Cells. List and origin of cells used throughout the *in vitro* experiments.

• Supplementary file 3. Mouse experiments. Details and ethics of mice used throughout the *in vivo* experiments.

• MDAR checklist

## Data availability

All data generated or analyzed during this study are included in the supporting file; Source Data files have been provided for Figures 1, 2, 4, 5 and 6. The materials generated during and/or analyzed during the current study are available from the corresponding authors on reasonable request.

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
