## [Editor Report]

The work provides a nice structure-function study providing clues as to how helix orientation can control transmission of signals across cell membranes. Moreover, the approach may find use in further studies exploring similar signalling involving other related membrane systems.

---

## [Decision Letter]

**Decision letter after peer review:**

Thank you for submitting your article "Constitutive Activation and Oncogenicity Are Mediated by Loss of Helical Structure at the Cytosolic Boundary of the Thrombopoietin Receptor" for consideration by *eLife*. Your article has been reviewed by 3 peer reviewers, and the evaluation has been overseen by a Reviewing Editor and Jonathan Cooper as the Senior Editor. The reviewers have opted to remain anonymous.

All reviewers are enthusiastic about the choice of problem and feel that the data presented, including the biological data, and the writing is of high quality. However, there is a strong feeling that additional data must be provided concerning the role and impact of W515 to further establish helix unwinding and its role in receptor activation. It is likely that new experiments will be necessary.

Essential revisions:

1) There is one major shortcoming in this manuscript, which concerns W515. It is known that mutation of W515 to any of 17 of the canonical amino acids, including Phe, is sufficient to trigger homodimerization and receptor activation. The authors present some evidence that the phenomenon behind this is that mutation of W515 to almost any other residues disrupts the helical secondary structure of the critical juxtamembrane segment, which promotes dimerization and receptor activation. What is puzzling is why a Trp at site 515 promotes helix formation, but nearly all other amino acids at this site disrupt helix formation. This strongly suggests the side chain of W515 must be interacting with another domain of the protein in the inactive state, in a manner that is responsible for how Trp stabilizes the juxtamembrane helix which is a central feature that helps define that state. This dangling missing piece of their mechanistic model should be resolved. Perhaps REDOR measurements could be performed to accomplish this goal.

2) While the solid state data is of very high quality, establishing clear differences between wild-type and mutants, there remains some concern about the unwinding of the helix with mutations, established on the basis of a single CO probe at a specific site. Additional experimental (NMR) confirmation of this effect, perhaps using additional nuclei or additional residues as probes, or preferably both, should be presented.

*Reviewer #3 (Recommendations for the authors):*

– Perhaps this work could be supported through additional predictions with alphafold or the like. Namely, if the predictions support the unfolding conclusion, that could be useful. Likewise, if the prediction is not supported by the experimental data, perhaps this would also hint at receptor clustering induced by TM domain mutations.

– Lines 168-169: "The secondary structure of the TpoR TM-JM peptides can be probed by solid-state NMR spectroscopy using peptides containing 13C-labeled amino acids at specific positions." I disagree with this statement. The chemical shift index and definition of secondary structures rely on consecutive residue observations just like Ramachandran angles, not a single chemical shift value.

– Line 188: "equivalent to L508 in mTpoR"; should be L505?

[Editors' note: further revisions were suggested prior to acceptance, as described below.]

Thank you for resubmitting your work entitled "Constitutive Activation and Oncogenicity Are Mediated by Loss of Helical Structure at the Cytosolic Boundary of the Thrombopoietin Receptor" for further consideration by *eLife*. Your revised article has been evaluated by Jonathan Cooper (Senior Editor) and a Reviewing Editor.

The manuscript has been improved but there are some remaining issues that need to be addressed, as outlined below:

The reviewers indicate that the SDS micelle data is problematic, likely resulting from the fact that SDS is not a good membrane mimic. Prior to publication this important point should be addressed.

*Reviewer #1 (Recommendations for the authors):*

The revised version of the manuscript has been significantly improved to address all previous concerns, including the provision of new data. I have no further recommendations to the authors. Nice paper!

*Reviewer #2 (Recommendations for the authors):*

While the authors addressed most of my critiques, the new solution NMR data included in the manuscript supporting the helical unwinding model are problematic. The authors choose a substandard membrane-mimicking system. SDS micelles are notorious for perturbing the structures of both soluble and membrane-bound proteins. Also, it is unclear if the chemical shift changes in micelles support the sparse chemical shift changes measured in lipid membranes. In Figure 3 suppl. 1, the W515K mutation seems to induce helicity for residues 501-509, while the remaining residues show slight unwinding. Again, these inconsistencies might be caused by the SDS micelles.

*Reviewer #3 (Recommendations for the authors):*

I have significant concerns about the new solution NMR data intended on addressing the reviewers' concern about the mechanism of activation.

1. The carbonyl chemical shift data in lipids and micelles (new data since original submission) on the same samples are in apparent contradiction. In figure 3 (panels e and f), WT – W515K would give a positive carbonyl chemical shift by ~2 ppm. However, in figure 3 supplement 1, the same difference gives a negative carbonyl chemical shift of ~0.5 ppm for the same residue (L512). Further, the trend from the solution NMR data indicates that the W515K mutant induces helicity (weakly) within the region around W515.

2. The solution NMR data are performed in SDS micelles, which is essentially an invalidated membrane mimic for studying membrane protein structures. Taken together with point #1, it is a significant cause for concern this environment is reporting on something biologically significant.

3. The authors included the Materials and methods with the figure legends. Within this is a materials and method section that includes "structural restraints and structure calculation" of the TM dimer; however, they do not show data or structures or discuss this in the text.

---

## [Author Response]

Essential revisions:1) There is one major shortcoming in this manuscript, which concerns W515. It is known that mutation of W515 to any of 17 of the canonical amino acids, including Phe, is sufficient to trigger homodimerization and receptor activation. The authors present some evidence that the phenomenon behind this is that mutation of W515 to almost any other residues disrupts the helical secondary structure of the critical juxtamembrane segment, which promotes dimerization and receptor activation. What is puzzling is why a Trp at site 515 promotes helix formation, but nearly all other amino acids at this site disrupt helix formation. This strongly suggests the side chain of W515 must be interacting with another domain of the protein in the inactive state, in a manner that is responsible for how Trp stabilizes the juxtamembrane helix which is a central feature that helps define that state. This dangling missing piece of their mechanistic model should be resolved. Perhaps REDOR measurements could be performed to accomplish this goal.

We agree with the reviewers that the mechanism by which Trp515 stabilizes the TM helix is central to the mechanism of activation. More broadly, our studies over the past decade have sought to address the importance of the entire RWQFP insert in the TM domain. Our working model for this sequence has been that cation-π interactions are central to the role of the Trp and the accompanying amino acids.

Arginine and tryptophan both are over-represented at the cytoplasmic TM-JM boundaries of membrane proteins. Arginine is positively charged and part of the “positive-inside” rule for membrane protein insertion. Arginine and lysine define the cytoplasmic ends of TM helices and prefer to be accessible to the water-exposed membrane surface. In contrast, tryptophan residues prefer hydrophobic head-group or membrane interior locations. A revealing aspect of the RWQFP motif is that the arginine and tryptophan are located at the membrane to cytosolic border. As a result, in order to accommodate arginine in a more water-inaccessible membrane environment, it interacts with the surface of the tryptophan indole ring. Partitioning of the RWQF sequence in a more water-inaccessible environment also drives the formation of helical secondary structure as an unpaired backbone C=O…NH in a hydrophobic environment is estimated to cost 3-6 kcal/mol of energy.

We have taken two approaches in respond to this essential criticism of the reviewers: one structural and one computational. Additional NMR data (structural approach) has been included in the supporting information (see response to point 2 below). Computational approaches provide a second way to address whether a cation–π interaction between Trp515 and the positively charged Arg514 is responsible for stabilizing the C-terminal TM helix. We have included a new supporting figure using Α-Fold 2.0 that probes the structural changes upon mutation of Trp515. In the wild-type receptor, Arg514 is predicted to form a cation–π interaction with Trp515. In the W515K mutant, the helical secondary structure in the RKQFP sequence is disrupted and Arg514 forms a new cation–π interaction with Trp529. Similar changes occur in other Trp515 mutants (e.g. W515A) highlighting the ability of Α-Fold to predict such interactions and the consequences of mutation. Overall, 15 out of 19 W515X mutants are predicted to be unfolded. Experimentally, 17 out of 19 mutations lead to activation. Importantly, W515C and W515P are the only two amino acid substitutions that do *not* cause constitutive activity experimentally (Defour, Chachoua, Pecquet, and Constantinescu, 2016). Computationally, these two sites do *not* predict helix unraveling. In short, the overall predictions of Α-Fold agree with the unique nature of tryptophan at position 515.

In addition, we have expanded the arguments supporting the potential role of cation–π interactions by adding a new section entitled “Unfolding of the RWQF α-helical motif is a common mechanism of receptor activation”.

These modifications are now in the revised manuscript starting with line 213:

“Our working model for the mechanism of activation in the wild-type or mutant receptors is that the RWQF motif is stabilized in the inactive state as an α-helix as a result of a cation-π interaction between R514 and W515. This interaction allows the RWQF sequence to partition into the more hydrophobic head-group region of the bilayer. Both Arg and Trp are over-represented at the cytoplasmic ends of TM helices (von Heijne, 1992), but whereas Arg prefers a water-accessible environment, Trp prefers to be buried in a more hydrophobic environment (Yau, Wimley, Gawrisch, and White, 1998). Since Arg and Trp are located at the border between membrane and cytosolic domains and Arg precedes Trp in the sequence, partitioning into the membrane head-group region results in a favorable interaction of the positive charge associated with the guanidinium group of the R514 side chain with the partial negative charge associated with the aromatic surface of the W515 side chain. Partitioning of the RWQF sequence into the more water-inaccessible environment drives the formation of helical secondary structure as an unpaired backbone C=O…NH in a hydrophobic environment is estimated to cost 6 kcal/mol of energy (Engelman, Steitz, and Goldman, 1986).

In this model, activation of the receptor results in or is caused by disruption of the R514-W515 cation-π interaction. In the W515 mutants, R514 is no longer stabilized in a membrane environment and the helix containing the RWQFP sequence unravels to allow the positively charged side chain to reach outside of the membrane. In the case of the Asn mutants and in the wild-type receptor with bound Tpo, dimerization of hTpoR (or rotation of the TM helices in mTpoR dimer), places W515 in the center of the helix-helix interface. The data suggest that a steric clash of the W515 side chains results in unraveling of the cytoplasmic end of the TM helix.

Computational and additional NMR data are provided in the supplementary figures to support the model of helix unraveling suggested by the solid-state NMR studies. Computationally, we used AlphaFold 2.0 (Jumper et al., 2021) calculations of hTpoR TM-JM peptides to predict the influence of all possible mutations at position 515 on the TM-JM helix structure. Remarkably, α-helix unraveling was predicted for 15 out of 20 possible amino acids at 515 (supplement 2 to Figure 3). Importantly, two of the mutations that are not predicted to cause helix unraveling are W515C and W515P. Experimentally, these two amino acid substitutions are the only ones that do not induce constitutive activity among all possible amino acid substitutions at W515 (Defour et al., 2016). Introducing a Trp at the preceding position 514 instead of R/K in W515K/R mutants reverses helix unfolding in AlphaFold simulations (supplement 3 to Figure 3). This result agrees with our previous data that the WRQFP mutant is inactive and is essentially monomeric (J. P. Defour et al., 2013).

Structurally, we have undertaken solution-NMR studies of the wild-type hTpoR TM-JM peptide and its W515K mutant. Relaxation measurements of the backbone ^15^N resonances show that W515K mutation leads to association of the TM helices, and that it induces upfield chemical shift changes in the RWQF sequence consistent with helix unraveling (supplement 1 to Figure 3).”

2) While the solid state data is of very high quality, establishing clear differences between wild-type and mutants, there remains some concern about the unwinding of the helix with mutations, established on the basis of a single CO probe at a specific site. Additional experimental (NMR) confirmation of this effect, perhaps using additional nuclei or additional residues as probes, or preferably both, should be presented.

Solid-state NMR spectroscopy has the advantage of allowing measurements of the TM-JM domains of the membrane receptors in membrane environments. The membrane bilayer can have a direct influence on protein structure and correspondingly on the mechanism of receptor activation. Two aspects of the TM-JM structures that are important in such systems are dimerization and interactions of JM residues with the lipid head-groups. We have shown in previous studies on the TpoR that peptides corresponding to the TM-JM region can reproduce the dimerization behavior observed with full receptors and receptor mutants (J.-P. Defour et al., 2013). In the TpoR, the TM-JM peptides provide clear links between dimerization and changes in the RWQFP region and switch region. That is, the “dimerization switch” appears to be contained in the residues in this limited region.

To address the reviewer concern that the NMR data reporting on a single ^13^C probe is too limited, we have included a comparison of wild-type and W515K TpoR constructs using solution NMR spectroscopy in Supplement 2 to Figure 3*.* The solution NMR studies on the wild-type TM-JM peptides and the W515K mutant show that mutation leads to dimerization (as reflection in a change in NMR correlation times), and also leads to a change in the chemical shifts in the RWQFP region consistent with the solid-state NMR data. The solution NMR studies have the advantage of probing the entire TM-JM structure, and provide complementary support to helix unraveling induced by dimerization in an active orientation of the TM-JM peptides. Our overall strategy has been to extend the solution NMR studies to larger TpoR constructs (i.e. constructs that include the cytoplasmic domains of the receptor), but these studies are beyond the scope of this manuscript.

The text has been changed at Line 213 as indicated above.

Reviewer #3 (Recommendations for the authors):– Perhaps this work could be supported through additional predictions with alphafold or the like. Namely, if the predictions support the unfolding conclusion, that could be useful. Likewise, if the prediction is not supported by the experimental data, perhaps this would also hint at receptor clustering induced by TM domain mutations.– Lines 168-169: "The secondary structure of the TpoR TM-JM peptides can be probed by solid-state NMR spectroscopy using peptides containing 13C-labeled amino acids at specific positions." I disagree with this statement. The chemical shift index and definition of secondary structures rely on consecutive residue observations just like Ramachandran angles, not a single chemical shift value.

We agree that additional data is required to define secondary structure. We have previously determined through FTIR measurements that the TM-JM peptides adopt α-helical structure without β-sheet or β-strand contributions. The helical structure is more difficult to deconvolute from random coil with this type of analysis. However, the ^13^C=O chemical shift is diagnostic of the helix-coil transition due to the large change in hydrogen-bonding. McDermott and co-workers showed a nice correlation between the sigma22 component of the chemical shift for ^13^C=O groups and hydrogen bonding strength derived from small molecule crystal structures and FTIR measurements (Gu, Zambrano, and McDermott, 1994). For this manuscript, we sought to present a single (simple) marker in a large number of mutants and to determine if there was a correlation with activity of the full-length receptor. In parallel studies we have been using solution NMR to obtain assignments and constraints on these peptides in detergent. We have included a comparison of the WT and W515K TM-JM peptides that arrives at the same conclusions as for the solid-state NMR studies. We have also reworded this section referenced above as follows.

Revised text added at page 8 line 183:

“To gain insight into the secondary structure of TpoR TM-JM peptides we combined solid state NMR and FTIR. We performed solid-state NMR spectroscopy using peptides containing ^13^C-labeled amino acids at specific positions. Assignments were aided by FTIR measurements of the amide I vibration which reveal that the TM-JM peptides are predominantly a-helical when reconstituted into membrane bilayers (J. P. Defour et al., 2013) without contributions from b-strand or b-sheet secondary structure. It is more challenging to distinguish random coil from α-helix by FTIR due to overlap of the vibrational bands. However, on the basis of the ^13^C NMR chemical shift of backbone carbonyl groups, we have previously shown that the α-helix of the TM domain extends into the JM region until at least F510, which is adjacent to P511 of the KWQFP motif (reference Staerk et al. 2006, Blood 107, 1864). The ^13^C=O NMR chemical shift is sensitive to secondary structure and occurs at ~175 ppm or higher when the ^13^C-labeled amino acid is within an α-helix due to direct hydrogen bonding to the i-4 NH group (Saitô, Tuzi, and Naito, 1998).”

– Line 188: "equivalent to L508 in mTpoR"; should be L505?

Indeed, we thank the reviewer for this correction. This has been corrected.

References

Bell, C. A., Tynan, J. A., Hart, K. C., Meyer, A. N., Robertson, S. C., and Donoghue, D. J. (2000). Rotational coupling of the transmembrane and kinase domains of the Neu receptor tyrosine kinase. *Mol Biol Cell, 11*(10), 3589-3599. Retrieved from <Go to ISI>://000089834800025

Braun, P., and von Heijne, G. (1999). The aromatic residues Trp and Phe have different effects on the positioning of a transmembrane helix in the microsomal membrane. *Biochemistry, 38*(30), 9778-9782. doi:doi:10.1021/bi990923a

Brooks, A. J., Dai, W., O'Mara, M. L., Abankwa, D., Chhabra, Y., Pelekanos, R. A.,... Waters, M. J. (2014). Mechanism of activation of protein kinase JAK2 by the growth hormone receptor. *Science, 344*(6185), 1249783. doi:10.1126/science.1249783

Defour, J.-P., Itaya, M., Gryshkova, V., Brett, I. C., Pecquet, C., Sato, T.,... Constantinescu, S. N. (2013). Tryptophan at the transmembrane-cytosolic junction modulates thrombopoietin receptor dimerization and activation. *Proceedings of the National Academy of Sciences of the United States of America, 110*(7), 2540-2545. doi:10.1073/pnas.1211560110

Defour, J. P., Chachoua, I., Pecquet, C., and Constantinescu, S. N. (2016). Oncogenic activation of MPL/thrombopoietin receptor by 17 mutations at W515: implications for myeloproliferative neoplasms. *Leukemia, 30*(5), 1214-1216. doi:10.1038/leu.2015.271

Defour, J. P., Itaya, M., Gryshkova, V., Brett, I. C., Pecquet, C., Sato, T.,... Constantinescu, S. N. (2013). Tryptophan at the transmembrane-cytosolic junction modulates thrombopoietin receptor dimerization and activation. *Proc Natl Acad Sci U S A, 110*(7), 2540-2545. doi:10.1073/pnas.1211560110

Engelman, D. M., Steitz, T. A., and Goldman, A. (1986). Identifying nonpolar transbilayer helices in amino acid sequences of membrane proteins. *Annual Review of Biophysics and Biophysical Chemistry, 15*, 321-353.

Gu, Z., Zambrano, R., and McDermott, A. (1994). Hydrogen Bonding of Carboxyl Groups in Solid-State Amino Acids and Peptides: Comparison of Carbon Chemical Shielding, Infrared Frequencies, and Structures. *Journal of the American Chemical Society, 116*(14), 6368-6372. doi:10.1021/ja00093a042

Hall, B. A., Armitage, J. P., and Sansom, M. S. P. (2011). Transmembrane helix dynamics of bacterial chemoreceptors supports a piston model of signalling. *PLoS Computational Biology, 7*(10), e1002204. doi:10.1371/journal.pcbi.1002204

Jumper, J., Evans, R., Pritzel, A., Green, T., Figurnov, M., Ronneberger, O.,... Hassabis, D. (2021). Highly accurate protein structure prediction with AlphaFold. *Nature*, 1-11. doi:10.1038/s41586-021-03819-2

Leroy, E., Defour, J. P., Sato, T., Dass, S., Gryshkova, V., Shwe, M. M.,... Smith, S. O. (2016). His499 Regulates Dimerization and Prevents Oncogenic Activation by Asparagine Mutations of the Human Thrombopoietin Receptor. *J Biol Chem, 291*(6), 2974-2987. doi:10.1074/jbc.M115.696534

Liu, W., Crocker, E., Constantinescu, S. N., and Smith, S. O. (2005). Helix packing and orientation in the transmembrane dimer of gp55-P of the spleen focus forming virus. *Biophysical Journal, 89*(2), 1194-1202. doi:doi:10.1529/biophysj.104.057844

McLaughlin, S., Smith, S. O., Hayman, M. J., and Murray, D. (2005). An electrostatic engine model for autoinhibition and activation of the epidermal growth factor receptor (EGFR/ErbB) family. *Journal of General Physiology, 126*(1), 41-53. doi:doi:10.1085/jgp.200509274

Pope, A. L., Sanchez-Reyes, O. B., South, K., Zaitseva, E., Ziliox, M., Vogel, R.,... Smith, S. O. (2020). A conserved proline hnge mediates helix dynamics and activation of rhodopsin. *Structure, 28*(9), 1004-1013. doi:10.1016/j.str.2020.05.004

Ren, Z., Ren, P. X., Balusu, R., and Yang, X. J. (2016). Transmembrane helices tilt, bend, slide, torque, and unwind between functional states of rhodopsin. *Sci Rep, 6*. doi:34129 10.1038/srep34129

Saitô, H., Tuzi, S., and Naito, A. (1998). Empirical Versus Non Empirical Evaluation Of Secondary Structure Of Fibrous And Membrane Proteins By Solid-State Nmr: A Practical Approach. *Annual Reports on NMR Spectroscopy, 36*, 79-121. doi:10.1016/s0066-4103(08)60006-x

Sato, T., Pallavi, P., Golebiewska, U., McLaughlin, S., and Smith, S. O. (2006). Structure of the membrane reconstituted transmembrane-juxtamembrane peptide EGFR(622-660) and its interaction with ca^2+^/calmodulin. *Biochemistry, 45*(42), 12704-12714. doi:doi:10.1021/bi061264m

Sato, T., Tang, T. C., Reubins, G., Fei, J. Z., Fujimoto, T., Kienlen-Campard, P.,... Smith, S. O. (2009). A helix-to-coil transition at the ε-cut site in the transmembrane dimer of the amyloid precursor protein is required for proteolysis. *Proceedings of the National Academy of Sciences of the United States of America, 106*(5), 1421-1426. doi:doi:10.1073/pnas.0812261106

Smith, S. O., Eilers, M., Song, D., Crocker, E., Ying, W. W., Groesbeek, M.,... Aimoto, S. (2002). Implications of threonine hydrogen bonding in the glycophorin A transmembrane helix dimer. *Biophysical Journal, 82*(5), 2476-2486. Retrieved from <Go to ISI>://000175259600017

Smith, S. O., Smith, C., Shekar, S., Peersen, O., Ziliox, M., and Aimoto, S. (2002). Transmembrane interactions in the activation of the Neu receptor tyrosine kinase. *Biochemistry, 41*(30), 9321-9332. doi:doi:10.1021/bi012117l

Tang, T. C., Hu, Y., Kienlen-Campard, P., El Haylani, L., Decock, M., Van Hees, J.,... Smith, S. O. (2014). Conformational changes induced by the A21G Flemish mutation in the amyloid precursor protein lead to increased Aβ production. *Structure, 22*(3), 387-396. doi:10.1016/j.str.2013.12.012

Tang, T. C., Kienlen-Campard, P., Hu, Y., Perrin, F., Opsomer, R., Octave, J. N.,... Smith, S. O. (2019). Influence of the familial Alzheimer's disease-associated T43I mutation on the transmembrane structure and γ-secretase processing of the C99 peptide. *J Biol Chem, 294*(15), 5854-5866. doi:10.1074/jbc.RA118.006061

von Heijne, G. (1992). Membrane-protein structure prediction – Hydrophobicity analysis and the positive-inside rule. *Journal of Molecular Biology, 225*(2), 487-494. doi:10.1016/0022-2836(92)90934-c

Yau, W. M., Wimley, W. C., Gawrisch, K., and White, S. H. (1998). The preference of tryptophan for membrane interfaces. *Biochemistry, 37*(42), 14713-14718. doi:doi:10.1021/bi980809c

Zhang, W. Y., Sato, T., and Smith, S. O. (2006). NMR spectroscopy of basic/aromatic amino acid clusters in membrane proteins. *Progress in Nuclear Magnetic Resonance Spectroscopy, 48*(4), 183-199. doi:doi:10.1016/j.pnmrs.2006.04.002

[Editors' note: further revisions were suggested prior to acceptance, as described below.]

The manuscript has been improved but there are some remaining issues that need to be addressed, as outlined below:The reviewers indicate that the SDS micelle data is problematic, likely resulting from the fact that SDS is not a good membrane mimic. Prior to publication this important point should be addressed.

There were two major concerns from the reviewers 2 and 3 concerning the solution NMR data that are now addressed and incorporated into the revised manuscript. The first was the inconsistency between the solution and solid-state NMR data. This came from our error in labeling panel "d" of Suppl Figure 1 of Figure 3, data in panel "d" are representing chemical shifts of W515K versus wild type (WT) TpoR, while in the legend and on the panel the label was incorrectly TpoR WT versus W515K. We apologize for this error. The second was that SDS is a detergent and a sub-standard membrane mimic. We have shown that SDS allowed us to reproduce the ability of W515K to induce dimerization of WT TpoR (which is a monomer). The behaviors of WT TpoR as monomer and TpoR W515K as dimer were established by us in previous papers both in membrane environment (Leroy, Defour et al. 2016) as well as in detergent micelles (dodecylphosphocholine, DPC) (Defour, Itaya et al. 2013). SDS allowed us to observe the same dimerization as in membrane environment, and allowed us to obtain higher resolution spectra than in DPC micelles. We then show that in dimerizing conditions W515K induces unraveling of the TpoR sequence encompassing residues 514-517. We thank the reviewer for the concern because this made us realize that we did not emphasize enough an important point: W515 or S505N mutations induce unraveling because they induce dimerization of the transmembrane domains. This has been shown by us in two previous publications using analytical ultracentrifugation, infrared spectroscopy, solid-state NMR and the environment was DMPC:DMPG, a more physiological environment than SDS. We stress this point now at page 10.

We added at Results page 8 the following sentence line 174:

"We have previously shown that the TM-JM peptides reconstituted into model membrane bilayers composed of DMPC:DMPG are able to replicate the dimerization behavior of the full TpoR wild-type and mutant receptors (21). The negatively charged DMPG provides a net negative charge to the membrane surface that mimics inner bilayer surface of native plasma membranes (see Methods)."

Reviewer #2 (Recommendations for the authors):While the authors addressed most of my critiques, the new solution NMR data included in the manuscript supporting the helical unwinding model are problematic. The authors choose a substandard membrane-mimicking system. SDS micelles are notorious for perturbing the structures of both soluble and membrane-bound proteins. Also, it is unclear if the chemical shift changes in micelles support the sparse chemical shift changes measured in lipid membranes. In Figure 3 suppl. 1, the W515K mutation seems to induce helicity for residues 501-509, while the remaining residues show slight unwinding. Again, these inconsistencies might be caused by the SDS micelles.

The plotted data in panel "d" of Supplemental Figure 1 of Figure 3 are the differences of W515K compared to the WT chemical shifts. The data are consistent between solution NMR and solid-state NMR in that both exhibit upfield chemical shifts in the W515K mutant compared to the WT sequence. The label of the panel and the figure legend have been revised as follows to clarify this point. We are sorry for the error/wording in our earlier version that led to the apparent discrepancy, which is not real.

Specifically, we have added the following sentences to the Figure Legend.

“The ability of the W515K peptide to dimerize in SDS micelles compared to the wild-type sequence mimics the observation of dimerization in membrane bilayers (21, 24).” (page 35, line 1000)

“The negative chemical shift differences indicate a lower frequency for the W515K mutant compared to wild-type. The upfield chemical shifts in the W515K mutant from Leu511 to the C-terminus are consistent with the carbonyl chemical shifts observed in the solid-state NMR resonances at Leu512 and are interpreted as unraveling of the C-terminus upon dimerization. The chemical shift difference at position 512 of -0.5 ppm in the solution NMR measurements is smaller than the difference in the solid-state NMR measurements of approximately -3 ppm.” (page 36, line 1003)

We also added at page 10, lane 239

“Structurally, we have undertaken solution-NMR studies in sodium dodecylsulfate (SDS) of the wild-type hTpoR TM-JM peptide and its W515K mutant. Relaxation measurements of the backbone ^15^N resonances show that W515K mutation leads to association of the TM helices (as observed in membrane bilayers), and that it induces upfield chemical shift changes in the RWQF sequence consistent with helix unraveling (supplement 1 to Figure 3). Similar results were obtained for the S505N mutant of TpoR (481-520) (45).”

Reviewer #3 (Recommendations for the authors):I have significant concerns about the new solution NMR data intended on addressing the reviewers' concern about the mechanism of activation.1. The carbonyl chemical shift data in lipids and micelles (new data since original submission) on the same samples are in apparent contradiction. In figure 3 (panels e and f), WT – W515K would give a positive carbonyl chemical shift by ~2 ppm. However, in figure 3 supplement 1, the same difference gives a negative carbonyl chemical shift of ~0.5 ppm for the same residue (L512). Further, the trend from the solution NMR data indicates that the W515K mutant induces helicity (weakly) within the region around W515.2. The solution NMR data are performed in SDS micelles, which is essentially an invalidated membrane mimic for studying membrane protein structures. Taken together with point #1, it is a significant cause for concern this environment is reporting on something biologically significant.

We expand below our arguments for the use of SDS in the solution NMR experiments as support for the solid-state NMR measurements. We recognize that detergent micelles do not have the same physical properties as membrane bilayers. Our structural studies on TpoR over the past decade have been almost exclusively in cell membranes or in model membrane bilayers, so we fully understand the concern. Despite their differences, both detergents and membrane bilayers induce helical structure in the hydrophobic transmembrane portion of single membrane-spanning sequences. The reason is that both detergents and membranes exclude water from the backbone amide groups of the sequence. In the absence of water, helix formation is favored via intra-helical hydrogen-bonding of these groups. Nevertheless, a key criterion for us as to whether detergents are able to “mimic” membrane bilayers is whether they are able to reproduce the ability of the transmembrane helices to associate in a biologically relevant manner, that is for example in the case of TpoR to reproduce the dimerization induced by W515K in TpoR TM domains that we reported with DMPC:DMPG and was validated by cell biological assays in our previous papers for the full length TpoR.

An important observation in this regard is that the human wild-type TpoR sequence is monomeric in cell membranes and dimerizes upon ligand binding (Leroy, Defour et al. 2016). Importantly, the receptor dimerizes due to the activating mutations at Ser505 and Trp515 (Leroy, Defour et al. 2016). As a result, there are basically two reasons for selecting SDS as the detergent for our solution NMR studies. The first is that SDS environment reproduces the dimerization behavior of the W515K mutant compared to the wild-type peptide/protein, which exists as a monomer. We had previously shown that both the W515K and S505N transmembrane sequences induced dimerization in membrane bilayers (DMPC:DMPG) and in detergent micelles (dodecylphosphocholine, DPC) (Defour, Itaya et al. 2013). That is, the wild-type TpoR sequence was observed to be monomeric under both detergent and membrane bilayer conditions, while TpoR sequences with activating mutations were dimeric under both conditions.

The second reason for selecting SDS is that SDS yields slightly higher-resolution spectra of both the monomeric wild-type peptide and the W515K peptide than DPC. For our studies on the TpoR peptides, we screened several different detergents (including isotropic bicelles formed from short and long chain lipids) and found that (perhaps surprisingly) SDS provided the highest resolution spectra exhibiting the expected dimerization of the W515K mutant.

Our original studies on the dimerization of transmembrane helices traces back to the work on the TM domain of glycophorin A, which was known to dimerize via its TM sequence. These studies were undertaken in both detergent micelles (MacKenzie, Prestegard et al. 1997) and in membrane bilayers (Smith, Song et al. 2001). The measurements yielded similar but not identical dimer interfaces. The major differences related to two characteristics of detergent micelles: they do not have defined planar surfaces (as in bilayers) and they are more permeable to water than bilayers. The result in the case of glycophorin A was that the crossing-angle of the helices was slightly larger and one of the residues within the interface (Thr87) formed different hydrogen bonding contacts (Smith, Eilers et al. 2002). In membranes, the Thr87 side chain formed intermolecular hydrogen bonds (not found in the detergent dimer structure) (Smith, Eilers et al. 2002). This subtle difference was attributed to the larger crossing angle and association of water with Thr87 in micelles. That is, the differences between detergents and membrane bilayers were not large enough to markedly influence the dimerization between TM helices.

We had not undertaken the solution NMR studies on TpoR in response to the previous review, we had performed these experiments before this manuscript was submitted. Rather these results are part of a series of wider studies investigating the structure and interactions of TpoR. We argued that if the structures of the TM dimer in solution and membranes were the same (or roughly the same), we could use the SDS system to investigate the structures of the folded intracellular domains. For the current studies, we selected a single ^13^C probe that could be used in a series of peptides that included the human wild-type sequence, the murine wild-type sequence, several asparagine mutants in both human and mouse, several Trp515 mutants, and alanine insertion mutants. For additional NMR support for unraveling, the comparison of only the wild-type human and W515K mutant (in a completely independent set of measurements) extends the probes to additional ^13^C backbone C=O groups encompassing the 514-517 human TpoR insertion sequence (RWQFP or RKQFP). The reasoning (in part) was that this (along with results from the Α-Fold calculations) would be more straightforward for the reader to understand.

We have revised the text to clarify the points above.

Page 10, line 239. *Structurally, we have undertaken solution-NMR studies in sodium dodecylsulfate (SDS) of the wild-type hTpoR TM-JM peptide and its W515K mutant. Relaxation measurements of the backbone ^15^N resonances show that W515K mutation leads to association of the TM helices (as observed in membrane bilayers), and that it induces upfield chemical shift changes in the RWQF sequence consistent with helix unraveling (supplement 1 to Figure 3). Similar results were obtained for the S505N mutant of TpoR (481-520) (45). We had previously shown that both the W515K and S505N transmembrane sequences induced dimerization in membrane bilayers (DMPC:DMPG) as well as in detergent micelles (dodecylphosphocholine, DPC) (Defour, Itaya et al. 2013). The solution NMR studies provide an independent probe of the structure of the region surrounding W515 upon helix dimerization. These studies show that the transmembrane domain of TpoR is helical in both the wild-type and mutant peptides and that dimerization induced by the W515K mutation results in unraveling of the helix in the RWQF insert region of the peptide.*

Finally reviewer 3 indicated that we included the methods for how the assignments were made in the solution NMR experiments with the figure legends.

3. The authors included the Materials and methods with the figure legends. Within this is a materials and method section that includes "structural restraints and structure calculation" of the TM dimer; however, they do not show data or structures or discuss this in the text.

These data were mainly to make the point that we could determine that the dimer interfaces in the solution samples were the same as in membranes. The current manuscript was not intended to be a structural study of the W515K dimer in detergents. These sections have now been abbreviated.

References

Brett, I. C. (2012). Transmembrane domain structure and function in the erythropoietin receptor. Ph.D. Ph.D. thesis, Stony Brook University.

Defour, J.-P., M. Itaya, V. Gryshkova, I. C. Brett, C. Pecquet, T. Sato, S. O. Smith and S. N. Constantinescu (2013). "Tryptophan at the transmembrane-cytosolic junction modulates thrombopoietin receptor dimerization and activation." Proceedings of the National Academy of Sciences of the United States of America 110(7): 2540-2545.

Kay, L. E., D. A. Torchia and A. Bax (1989). "Backbone dynamics of proteins as studied by 15N inverse detected heteronuclear NMR-spectroscopy: Application to staphylococcal nuclease." Biochemistry 28(23): 8972-8979.

Leroy E., J.-P. Defour, T. Sato, S. Dass, V. Gryshkova, S.M. Marlar, J. Staerk, S.N. Constantinescu and S.O. Smith. (2016). "His499 regulates dimerization and prevents oncogenic activation by asparagine mutations of the human thrombopoietin receptor." Journal of Biological Chemistry 291(6): 2974-87.

MacKenzie, K. R., J. H. Prestegard and D. M. Engelman (1997). "A transmembrane helix dimer: Structure and implications." Science 276(5309): 131-133.

Smith, S. O., M. Eilers, D. Song, E. Crocker, W. W. Ying, M. Groesbeek, G. Metz, M. Ziliox and S. Aimoto (2002). "Implications of threonine hydrogen bonding in the glycophorin A transmembrane helix dimer." Biophysical Journal 82(5): 2476-2486.

Smith, S. O., D. Song, S. Shekar, M. Groesbeek, M. Ziliox and S. Aimoto (2001). "Structure of the transmembrane dimer interface of glycophorin A in membrane bilayers." Biochemistry 40(22): 6553-6558.